# TopoDistill: Distilling Global System Topology for Causal Discovery in Multivariate Time Series

Zehao Liu [1]   Pengfei Jiao [2]   Yuhan Wu [3]   Jianqi Yang [4,5]   Yuyu Yin[*] [1]

## Abstract

Although causal discovery from multivariate time series is widely used, it remains challenging under noise. Convergent cross mapping (CCM) infers causality by reconstructing shadow manifolds via time-delay embedding (TDE) and evaluating cross-map skill between manifolds. Despite Takens' theorem guarantees in ideal settings, TDE effectively attempts to recover system state from a single noisy view, often yielding geometrically degraded manifolds and unreliable distance-based neighborhoods, which in turn weakens causal identification. We propose TopoDistill, a topology-informed knowledge distillation framework that improves univariate shadow-manifold reconstruction by aligning local neighborhood structure to a multivariate system representation. A global embedder trained on multivariate observations captures a global attractor representation, while a delay embedder is distilled to produce embeddings whose neighborhood distributions match the global topology. This cross-view alignment yields smoother and more reliable neighborhoods, improving cross mapping under noise while maintaining specificity against spurious correlations. Theoretical analysis and experimental results demonstrate that our method enables effective causal discovery.

[1]School of Computer Science, Hangzhou Dianzi University, Hangzhou, China [2]School of Cyberspace, Hangzhou Dianzi University, Hangzhou, China [3]College of Computer Science and Technology, Zhejiang University, Hangzhou, China [4]School of Engineering, Westlake University, Hangzhou, China [5]Institute of Advanced Technology, Westlake Institute for Advanced Study, Hangzhou, China. Correspondence to: Yuyu Yin[*] <yinyuyu@hdu.edu.cn>.

*Proceedings of the 43rd International Conference on Machine Learning*, Seoul, South Korea. PMLR 306, 2026. Copyright 2026 by the author(s).

## 1. Introduction

Causal discovery from multivariate time series (MTS) is of significant importance across numerous scientific domains, such as healthcare, biology, and Earth sciences (Brown et al., 2025; Lagemann et al., 2023; Runge et al., 2023). By uncovering the causal relationships within MTS, we can gain deep insights into the underlying mechanisms of the systems under study and guide the design of interpretable and robust data-analytic models(Gong et al., 2024). The central challenge in this research lies in distinguishing genuine causal relationships from spurious correlations in systems that are inherently multivariate and nonlinear.

Multivariate causal discovery has been extensively studied. As one of the earliest systematic frameworks, Granger causality (Granger, 1969) detects causal relationships by testing the direction of information flow between variables in a prediction setting. However, it relies on a separability assumption. This assumes that causal effects can be decomposed into independent contributions. This assumption may be violated in coupled nonlinear systems.

Convergent cross mapping (CCM) (Sugihara et al., 2012) offers a complementary dynamical-systems-based perspective that does not rely on the separability assumption. Building on Takens' theorem (Takens, 1981), CCM reconstructs shadow manifolds of the system via delay embedding and identifies causal relationships by testing whether the historical information of one variable can reconstruct another. Owing to its ability to detect weak coupling in chaotic systems, CCM has been widely applied in a variety of complex settings.

Although Takens' theorem guarantees the topological equivalence of the reconstructed attractor in the noise-free setting, standard time-delay embedding (TDE) faces challenges under complex noise. TDE applies a rigid linear transformation to the input series, so any measurement noise is directly propagated into the embedding space. This yields a reconstructed shadow manifold that is geometrically rough and distorted, manifested as pronounced jitter in phase-space trajectories (Fig. 1a). Because CCM relies on distances on the shadow manifold for prediction, such geometric degradation compromises predictive reliability. In particular, Euclidean

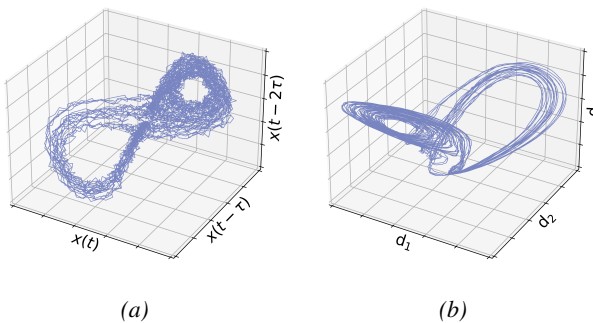

*(a)*            *(b)*

*Figure 1.* Visualization of the shadow manifold reconstructed from the $x$ series of the noise-corrupted Lorenz attractor. (a) With TDE, the trajectory exhibits severe jitter, disrupting the original neighborhood structure. (b) With our method, the manifold is smoother and the local neighborhoods are more clearly preserved.

distances on a noise-corrupted manifold may not faithfully reflect true dynamical proximity, thereby weakening causal discovery.

To address this limitation, we propose shifting the embedding paradigm from a rigid transformation to a learnable, topology-guided mapping. Our central motivation is that, compared with time-delay embeddings of a single series, snapshots of multivariate time series provide a more comprehensive description of the system state. In CCM, single-series embedding is essentially an attempt to recover the full system state from incomplete information, whereas multivariate observations naturally contain much of this information. Therefore, when reconstructing a shadow attractor from a univariate series is challenging, one can guide the embedder by aligning the reconstructed attractor with the local topology of the original system attractor, enabling more faithful shadow-attractor reconstruction from noisy data(Fig. 1b).

Notably, this framework does not aim to incorporate multivariate information into causal discovery, as doing so would violate CCM's fundamental premise that the shadow manifold is reconstructed solely from univariate observations. Instead, system-level information is used only as a supervisory signal, guiding the embedder to learn how to use limited observations to produce an embedding that is as faithful and minimally distorted as possible.

Specifically, we propose TopoDistill, which employs neural networks as a global embedder and a delay embedder to construct the system manifold and the univariate shadow manifold, respectively. The global embedder is trained on multivariate snapshots under a one-step-ahead prediction paradigm to capture system-level dynamics. We then adopt a knowledge distillation scheme to align the local topology of the shadow manifold with that of the system manifold—that is, maintaining the relative point ordering within

local neighborhoods to reflect true dynamical proximity rather than noise—induced spurious proximity. For the delay embedder, we further introduce a contrastive learning objective to explicitly enforce temporal continuity and to ensure that the shadow manifold is sufficiently unfolded.

Given that CCM is highly sensitive to shadow-manifold quality, we design an early-stopping criterion based on manifold smoothness to mitigate overfitting during training. At test time, embeddings produced by the delay embedder replace TDE and are fed into CCM for causal discovery.

Our contributions are summarized as follows:

- We propose a knowledge distillation framework that leverages system-level topological information to guide the training of a delay embedder, yielding robust shadow manifolds for CCM.

- We introduce a contrastive learning objective to enforce temporal continuity and sufficient unfolding of the shadow manifold, and develop a manifold-smoothness–based early-stopping criterion to ensure that the learned representations are well suited for causal discovery.

- We validate the effectiveness of our method on both synthetic and real-world datasets.

## 2. Related Works

**Discovering causal relations from MTS.** Existing approaches can be broadly categorized into four classes, including constraint-based methods, score-based methods, functional causal model based methods, and Granger causality based methods (Gong et al., 2024). The PC algorithm (Spirtes et al., 2000) and PCMCI (Runge et al., 2019) fall under constraint-based approaches and rely on statistical tests of conditional independence. DYNOTEARS (Pamfil et al., 2020) is a score-based method that captures linear relationships from time series data via continuous optimization. As a representative functional causal model based method, VAR-LiNGAM (Hyvärinen et al., 2008) identifies time lagged and instantaneous causal directions among variables within the vector autoregressive framework by leveraging non Gaussianity and the linear acyclic assumption.

Granger causality (Granger, 1969) based frameworks were initially limited to linear causal relationships. In recent years, many studies (Montalto et al., 2015; Tank et al., 2021; Marcinkevičs & Vogt, 2021) have extended this framework to nonlinear causality using neural networks. CUTS (Cheng et al., 2023) addresses time series with missing values, while CUTS+ (Cheng et al., 2024) further improves performance in high dimensional causal discovery. Several works have

proposed causal discovery methods tailored to specific time series tasks, such as forecasting (Li et al., 2023) and anomaly detection (Liu et al., 2025).

**CCM-based methods.** CCM (Sugihara et al., 2012) is a nonlinear causal discovery method based on state-space reconstruction and is particularly well suited to deterministic chaotic systems. A substantial body of work has improved the original CCM from different points of view. Spatial CCM (Clark et al., 2015) enables causal discovery from short time series by incorporating dewdrop regression. Ye et al. (Ye et al., 2015) incorporate time delays to identify causal relationships with different delays. Mønster et al. (Mønster et al., 2017) achieve CCM-based causal discovery under strong coupling by performing controlled noise injection in strongly coupled systems. Feng et al. (Feng et al., 2019) propose combining deep Gaussian processes to learn higher-quality attractor manifolds. Leng et al. (Leng et al., 2020) leverage partial correlation to better distinguish indirect from direct causality. To improve robustness to missing values, Latent CCM (De Brouwer et al., 2020) uses neural ODEs to generate continuous embeddings. The more recent study (Zhang et al., 2025) extends CCM to multivariate settings, using multivariate embeddings to prune indirect causal links and more effectively disentangle indirect causal relationships.

# 3. Preliminaries

## 3.1. Problem Definition

This paper focuses on causal discovery from multivariate time series. A multivariate time series can be represented as a sequence of time points: $\{\mathbf{x}_1, \mathbf{x}_2, ..., \mathbf{x}_T\}$, where $\mathbf{x}_t = \{(x_t^{(1)} x_t^{(2)} ... x_t^{(N)})^\top\}$. $T$ and $N$ denote the sequence length and the number of variables, respectively. Assume that the data-generating process is defined by the following structural equation model:

$$x_t^{(i)} := f_i\big(\mathrm{Pa}(x_t^{(i)}), u_t^{(i)}\big), \quad i = 1, \ldots, N, \quad (1)$$

where $\mathrm{Pa}(\mathrm{x}_\mathrm{t}^{(i)})$ denotes the set of causal parents of variable $i$, $u_t^{(i)}$ is an independent noise term, and $f_i$ is the structural equation that generates the $i$-th time series. The goal is to produce a binary causal adjacency matrix $A \in \mathbb{Z}^{N \times N}$: if variable $j$ belongs to the causal parent set of variable $i$, then $A_{ij} = 1$; otherwise, $A_{ij} = 0$. Following common practice (Zhang et al., 2025), we allow the model to produce a real-valued (continuous) matrix, which is then binarized using an empirical threshold.

## 3.2. Takens' Theorem and State-Space Reconstruction

The Takens' theorem exploits this idea by 'extending the dimension' of an observation signal by considering lagged copies of the signal as new observations (Sauer et al., 1991):

**Theorem 3.1** (Takens' Theorem). *Let $\mathbf{X}_{t+1} = F_{\mathbf{X}}(\mathbf{X}_t)$ be an autonomous deterministic system on a $D$-dimensional state space with one-dimensional observation $x = f(\mathbf{X}_t)$ and suppose that there is an attractor $A_{att} \subset \mathbb{R}^D$ such that $\mathbf{X}_t \in A_{att}$. Let $d$ and $\tau$ be fixed positive integers where $d > 2\dim(A_{att})$. Then the $d$-dimensional delay embedding of $x$, defined by*

$$m_a^t \triangleq \big[\, x_{t-(d-1)\tau} \;\cdots\; x_{t-\tau} \; x_t \,\big]^\top \in \mathbb{R}^d$$

*is a smooth embedding $A_{att} \to \mathbb{R}^d$ for almost-every $F_{\mathbf{X}}$ and $f$.*

The embedding method established by Takens' theorem is known as time-delay embedding (TDE). The attractor under diffeomorphism is called a shadow manifold, denoted by $\mathcal{M}_x$. Reconstructing the latent state $\mathbf{X}_t$ from observations is called state-space reconstruction (Butler et al., 2023).

## 3.3. Convergent Cross Mapping

CCM infers causal relationships and their direction by assessing whether the state of one variable can be cross-mapped from the delay-coordinate reconstruction of another variable.

A key assumption is that information about the cause is encoded in the history of the effect. Accordingly, the state of the causal driver can, in principle, be partially inferred from the TDE of the response variable. Because TDE suffers from the curse of dimensionality in high-dimensional settings, CCM typically allows the embedding dimension to deviate from the strict requirements of Takens' theorem.

Specifically, consider a deterministic dynamical system with two coupled variables, $\mathbf{X}$ and $\mathbf{Y}$. Under suitable genericity and embedding conditions, delay-coordinate reconstructions of $\mathbf{X}$ and $\mathbf{Y}$ give rise to shadow manifolds, $\mathcal{M}_x$, and $\mathcal{M}_y$, that capture the relevant underlying attractor structure. If $\mathbf{X}$ unidirectionally drives $\mathbf{Y}$, then the history of $\mathbf{Y}$ contains information about both variables, whereas the history of $\mathbf{X}$ generally does not contain comparable information about $\mathbf{Y}$. Accordingly, the state of $\mathbf{X}$ can in principle be estimated from $\mathcal{M}_y$, whereas recovering $\mathbf{Y}$ from $\mathcal{M}_x$ is expected to be substantially less successful in the absence of reverse causation.

CCM quantifies this asymmetry by performing cross mapping between $\mathcal{M}_x$ and $\mathcal{M}_y$ and evaluating the agreement between the cross-mapped estimates and the observed values, typically using the Pearson correlation coefficient. If $\mathbf{X}$ causally influences $\mathbf{Y}$, the cross-map skill from $\mathcal{M}_y$ to $\mathbf{X}$ should increase with library size and eventually level off, a phenomenon referred to as convergence. Causal inference is therefore based on the magnitude, directionality, and convergence of the cross-map skill. However, when the coupling

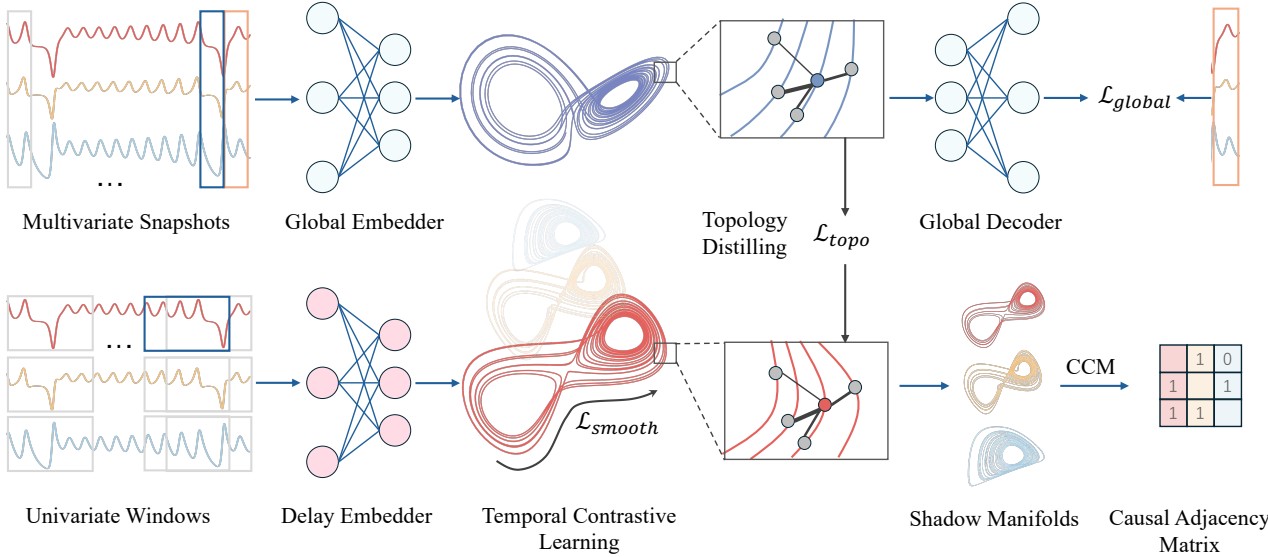

*Figure 2.* Overview of TopoDistill. A global embedder is first trained on multivariate observations to model the system's dynamical evolution. The neighborhood structure on the system attractor is then used as a supervisory signal, and knowledge distillation guides the delay embedder to better exploit information from univariate time windows. In addition, a temporal contrastive regularization term is introduced to enforce temporal consistency of the shadow manifold and to prevent pathological folding. The resulting shadow manifolds are subsequently used for CCM-based causal discovery.

is very strong, the two variables may approach synchronization, reducing the directional asymmetry on which CCM relies and thereby impairing causal identification. CCM is therefore generally most reliable under weak-to-moderate coupling.

CCM prediction relies on identifying nearest neighbors in the embedding space:

$$\hat{x}_t^{(j)} = \sum_{t_n \in \mathcal{N}_k(t)} w_n x_{t_n}^{(j)}, \quad w_n \propto \exp\left(-\|\mathbf{z}_t^{(i)} - \mathbf{z}_{t_n}^{(i)}\|/\epsilon\right), \tag{2}$$

where $\mathcal{N}_k(t)$ denotes the $k$ nearest neighbors of $\mathbf{z}_t^{(i)}$. The quality of these predictions and the reliability of the corresponding causal inference critically depend on whether the identified neighbors truly reflect dynamical similarity or spurious proximity induced by noise.

## 4. Methodology

TopoDistill first trains a global embedder on multivariate data: it takes a multivariate snapshot $\mathbf{x}_t$ as input and uses a global decoder to predict the next snapshot $\mathbf{x}_{t+1}$. The delay embedder takes a window $x_{t-\tau+1:t}^{(i)} = [x_{t-\tau+1}^{(i)}, ..., x_{t-1}^{(i)}, x_t^{(i)}]$ from a single time series as input and is trained to reconstruct the shadow manifold $\mathcal{M}_i$. The reconstructed shadow manifold is then used for CCM-based causal discovery. An overview of the model is shown in Fig 2.

### 4.1. Global Embedder

The global embedder aims to construct a reference manifold that captures the intrinsic dynamics of the system from multivariate observations. This reference manifold serves as a topological template that guides the reconstruction of shadow manifolds from univariate time series, ensuring that the learned embeddings preserve the underlying dynamical structure rather than merely compressing the raw observations. We employ a neural encoder $f_g : \mathbb{R}^n \to \mathbb{R}^{D_g}$ to map the observations to a latent representation:

$$\mathbf{z}_t^g = f_g(\mathbf{x}_t), \tag{3}$$

where $D_g$ is the embedding dimension. To ensure that $\mathbf{z}_t^g$ captures the dynamics of the system, we jointly train a decoder $f_{\text{dec}}$ under a one-step ahead prediction paradigm in multivariate snapshots. To ensure sufficient representational capacity, $f_g$ is instantiated as a multilayer perceptron. To encourage the system dynamics to be captured primarily by the global encoder rather than the global decoder, and motivated by a finite-dimensional Koopman approximation, $f_{\text{dec}}$ is chosen to be a single linear layer. The loss function is defined as:

$$\mathcal{L}_{\text{global}} = \mathbb{E}_t\left[\|\mathbf{x}_{t+1} - f_{\text{dec}}(\mathbf{z}_t^g)\|^2\right]. \tag{4}$$

By optimizing this objective, $f_g$ learns to approximate a set of Koopman observables $\phi = [\phi_1, \ldots, \phi_{D_g}]^\top$, yielding an embedding space $z_t = f_g(\mathbf{x_t}) \approx \phi(\mathbf{x_t})$. The embedding space $z$ preserves the essential characteristics of the system

dynamics, namely the locally linear structure of the manifold. Consequently, the topology derived from $z$ faithfully reflects the true underlying dynamics of the system.

Crucially, the prediction objective equips $f_g$ with an implicit denoising effect. Consider noisy observations $\tilde{\mathbf{x}}_t = \mathbf{x}_t + \boldsymbol{\epsilon}_t$, where $\boldsymbol{\epsilon}_t$ denotes measurement noise that is independent of future states. Because the loss penalizes only the prediction error of $\mathbf{x}_{t+1}$, the encoder is driven to extract features that maximize $I(\mathbf{z}_t^g; \mathbf{x}_{t+1})$ while suppressing components that are uninformative about future evolution. Consequently, at optimality,

$$f_g(\tilde{\mathbf{x}}_t) \approx f_g(\mathbf{x}_t) + \mathcal{O}(\|\boldsymbol{\epsilon}_t\|), \qquad (5)$$

and the perturbation term remains small when $\boldsymbol{\epsilon}_t \perp \mathbf{x}_{t+1}$. This property prevents noise-induced distortions in the raw observations from propagating into the embedding space, yielding a clean topological template that is robust to measurement corruption. A more detailed description is provided in Appendix A.

## 4.2. Delay Embedder

The rigid transformation in TDE inherits measurement noise, which can distort the geometric structure of local neighborhoods. Therefore, we replace TDE with a learnable neural network $f_s : \mathbb{R}^\tau \to \mathbb{R}^{D_s}$. Here $\tau$ is the length of the temporal window, and $D_s$ is the dimension of shadow embedding. The embedding is configured as:

$$\mathbf{z}_t^{(i)} = f_s^{(i)}(x_{t-\tau+1:t}^{(i)}), \qquad (6)$$

where $x_{t-\tau+1:t}^{(i)}$ denotes the temporal window of length $\tau$. The delay embedder is designed to be shared across all sequences. In principle, the architecture of the delay embedder can be generic and instantiated with any neural component suitable for time-series modeling. For simplicity, a TCN or an MLP is adopted as the delay embedder. Moreover, certain activation functions (e.g., ReLU), while introducing nonlinearity, may induce abrupt truncations in parts of the manifold, causing local regions to collapse. Therefore, the hyperbolic tangent function is used as the nonlinear activation.

## 4.3. Topology-Guided Distillation

A key property of a diffeomorphism is a weakened form of local isometry, meaning that: If $\phi : \mathcal{M}_i \to \mathcal{M}_j$ is a diffeomorphism, then for any point $p \in \mathcal{M}_i$ and its neighborhood $\mathcal{N}(p)$, the relative ordering of points within the neighborhood remains unchanged. This enables us to characterize the local topology via conditional probability distributions (Appendix B), analogous to the approach employed in t-SNE:

$$P_\mathcal{M}(v \mid u) = \frac{\exp(-\|m_u - m_v\|^2/\sigma^2)}{\sum_{k \neq u} \exp(-\|m_u - m_k\|^2/\sigma^2)}, \qquad (7)$$

where $m_i$ denotes the coordinate of the point $i$ in the manifold and $\sigma$ controls the neighborhood scale. Although the true system manifold is not directly observable, the global embedding induced by multivariate time series is closer to the underlying topology than that obtained from a single-sequence window.

The local topology of the global embedding $\{\mathbf{z}_t^g\}_{t=1}^T$ is used as the teacher distribution, characterizing the global manifold $\mathcal{M}_g$:

$$p_{t'|t}^g = \frac{\exp(-\|\mathbf{z}_t^g - \mathbf{z}_{t'}^g\|^2/T_{\text{emp}})}{\sum_{k \neq t} \exp(-\|\mathbf{z}_t^g - \mathbf{z}_k^g\|^2/T_{\text{emp}})}, \qquad (8)$$

where $T_{\text{emp}}$ denotes the temperature parameter. The local topology induced by the embedding $\{\mathbf{z}_t^{(i)}\}_{t=1}^T$ produced by the delay embedder is treated as the student distribution, characterizing the shadow manifold $\mathcal{M}_s^{(i)}$:

$$q_{t'|t}^{(i)} = \frac{\exp\left(-\|\mathbf{z}_t^{(i)} - \mathbf{z}_{t'}^{(i)}\|^2/T_{\text{emp}}\right)}{\sum_{k \neq t} \exp\left(-\|\mathbf{z}_t^{(i)} - \mathbf{z}_k^{(i)}\|^2/T_{\text{emp}}\right)}. \qquad (9)$$

If the shadow manifold is diffeomorphic to the system manifold, their local neighborhoods should exhibit consistent relative structures. Points that are neighbors on $\mathcal{M}_g$ should also be neighbors on $\mathcal{M}_s^{(i)}$, and vice versa. We enforce this via knowledge distillation, minimizing the Kullback-Leibler divergence:

$$\mathcal{L}_{\text{topo}}^{(i)} = \text{KL}(P^g \| Q^{(i)}) = \sum_{t=1}^T \sum_{t' \neq t} p_{t'|t}^g \log \frac{p_{t'|t}^g}{q_{t'|t}^{(i)}}. \qquad (10)$$

The total topology alignment loss aggregates over all $N$ time series:

$$\mathcal{L}_{\text{topo}} = \sum_{i=1}^N \mathcal{L}_{\text{topo}}^{(i)}. \qquad (11)$$

By minimizing $\mathcal{L}_{\text{topo}}$, the student encoder seeks to preserve topological consistency. Since the teacher distribution $p_{t'|t}^g$ is derived from $\mathbf{z}_t^g$, and $\mathbf{z}_t^g$ has been denoised through the predictive objective of the global embedder, aligning with $p_{t'|t}^g$ enables the student delay embedder $f_s^{(i)}$ to learn a distance metric that captures dynamical proximity. (Appendix C).

## 4.4. Temporal Contrastive Learning

Although $\mathcal{L}_{\text{topo}}$ enforces the neighborhood structure, it does not explicitly constrain the *temporal continuity* of the embedded trajectory. To prevent discontinuous jumps that could violate the manifold smoothness required for CCM, we augment the loss with a temporal contrastive term.

For each anchor embedding $\mathbf{z}_t^{(i)}$ at time $t$ in series $i$, we define positive samples as $\mathcal{P}(t, i) = \{t' : |t' - t| \leq$

$\Delta t_{\text{pos}}, t' \neq t\}$, where $\Delta t_{\text{pos}}$ is the threshold that defines the temporal window for positive samples. These represent states on the same trajectory segment that should be close in the embedding space. Negative samples are defined as $\mathcal{N}(t, i) = \{(t', j) : |t' - t| > \Delta t_{\text{neg}}\}$, where $\Delta t_{\text{neg}}$ is the threshold that defines the temporal window for negative samples.

We formulate the temporal contrastive loss using the InfoNCE framework (Oord et al., 2018). The similarity score is defined as:

$$s((t, i), (t', j)) \triangleq \exp\left(\frac{\text{sim}(\mathbf{z}_t^{(i)}, \mathbf{z}_{t'}^{(j)})}{T_{\text{emp}}}\right), \quad (12)$$

where $\text{sim}(\mathbf{z}_a, \mathbf{z}_b) = \mathbf{z}_a^\top \mathbf{z}_b / (\|\mathbf{z}_a\|\|\mathbf{z}_b\|)$ denotes cosine similarity, and $T_{\text{emp}} > 0$ is a temperature parameter controlling the concentration of the distribution. The temporal contrastive loss is then given by

$$\mathcal{L}_{\text{smooth}}^{(i)} = -\frac{1}{T} \sum_{t=1}^{T} \log \frac{\sum_{t' \in \mathcal{P}(t,i)} s((t, i), (t', i))}{\sum_{(t',j) \in \mathcal{P}(t,i) \cup \mathcal{N}(t,i)} s((t, i), (t', j))}. \quad (13)$$

The total contrastive loss aggregates over all series: $\mathcal{L}_{\text{smooth}} = \sum_{i=1}^{n} \mathcal{L}_{\text{smooth}}^{(i)}$. By minimizing $\mathcal{L}_{\text{smooth}}$, we encourage $z_t^{(i)}$ and $z_{t+1}^{(i)}$ to be close in the embedding space, while the negative-sample term pushes temporally distant states apart and prevents the degenerate solution in which all embeddings collapse to a single point. This ensures that the shadow manifold $\mathcal{M}_s^{(i)}$ does not degenerate, thereby preserving its diffeomorphic relationship with the attractor.

### 4.5. Training Strategies

Each dataset is split into a training set and a test set. The training set is used to train the model's neural-network components in a self-supervised manner, whereas CCM-based causal discovery is performed on the test set.

The teacher distribution produced by an insufficiently trained global embedder may deviate from the true neighborhood structure. To ensure stable training overall, we adopt a two-stage training strategy. In the first stage, we train the global embedder using only $\mathcal{L}_{\text{global}}$. In the second stage, the global encoder is frozen. The delay embedder $f_s$ is trained by jointly minimizing the topology alignment loss and the temporal contrastive loss:

$$\mathcal{L}_s = \lambda \mathcal{L}_{\text{topo}} + (1 - \lambda) \mathcal{L}_{\text{smooth}}, \quad (14)$$

where $\lambda$ is a hyperparameter balancing the two objectives.

The $\mathcal{L}_{\text{topo}}$ aims to align the topology of the shadow manifolds across all sequences with that of the underlying system manifold. However, a key issue is that a shadow manifold does not contain complete information about the system

attractor. As a result, even under ideal conditions, the topological loss generally cannot converge to zero, because a delay embedder cannot recover the full system topology from finite information in a single time series. Without proper control of training, forcing the topological loss to converge may lead to severe overfitting on the training data. In effect, the delay embedder fits spurious information from noise to match the teacher distribution as closely as possible, causing the shadow manifold to become increasingly rough. Therefore, we design an early-stopping strategy based on manifold smoothness, using the indicator:

$$S_m = \sum_i \sum_t \left\| \mathbf{z}_{t+1}^{(i)} - \mathbf{z}_t^{(i)} \right\|^2. \quad (15)$$

CCM requires the manifold to be continuous and smooth. If this value becomes too large, it indicates that the embedding is fitting noise. We therefore monitor this indicator dynamically during training and stop training when $S_m$ on training set begins to increase beyond a predefined patience threshold.

A more detailed discussion of the manifold smoothness indicator is provided in Appendix D.

### 4.6. CCM Scoring

To test whether $X$ drives $Y$, CCM constructs a shadow manifold from the time series of $Y$. For a given library size $L$, the *library* is the set of embedded points used as the reference database, while the *query* is the held-out embedded point $z_{\text{query}}$ at the target time $t$ (excluded from the library for cross-validation). For each query $\mathbf{y}_t$, CCM finds its $k_{\text{nn}}$ nearest neighbors in the library, maps their time indices back to $X$, and estimates $x_t$ by a distance-weighted average of $\{x_{t_i}\}$. Evidence for a causal influence $X \to Y$ is obtained if the correlation between $\hat{x}_t$ and $x_t$ increases and converges as $L$ grows.

Given a query embedding $\mathbf{z}_{\text{query}} \in \mathbb{R}^{D_s}$ and a library $\{\mathbf{z}_t, x_t\}_{n=t}^{L}$, we first compute pairwise distances in the embedding space:

$$d_t = \|\mathbf{z}_{\text{query}} - \mathbf{z}_t\|_2, \quad t = 1, \dots, L, \quad (16)$$

where $L$ is the length of the library. Select the $k_{\text{nn}}$ nearest neighbors: $\mathcal{N}_k = \{\arg\min_t d_t\}$. To emphasize closer neighbors, we use an exponential kernel with adaptive bandwidth:

$$w_t = \exp\left(-\frac{d_t}{d_0}\right), \quad d_0 = \min_{t \in \mathcal{N}_k} d_t, \quad (17)$$

where $d_0$ normalizes the distances by the closest neighbor, ensuring scale invariance. The predicted value is:

$$\hat{x}_{\text{query}} = \frac{\sum_{n \in \mathcal{N}_k} w_n x_n}{\sum_{n \in \mathcal{N}_k} w_n}. \quad (18)$$

*Table 1.* Performance of multivariate time-series causal discovery on four benchmark datasets, evaluated using AUC-ROC, AUC-PR, and SHD. The best results are highlighted in bold, and the second-best results are underlined.

| Dataset | Double Pendulum | | | UUMC | | | fMRI | | | Causal Rivers | | |
|---|---|---|---|---|---|---|---|---|---|---|---|---|
| Metric | ROC↑ | PR↑ | SHD↓ | ROC↑ | PR↑ | SHD↓ | ROC↑ | PR↑ | SHD↓ | ROC↑ | PR↑ | SHD↓ |
| CCM | 0.875 | 0.833 | 0.167 | 0.760 | 0.436 | 0.250 | 0.712 | 0.335 | 0.111 | 0.594 | 0.324 | 0.200 |
| | ±0.000 | ±0.000 | ±0.000 | ±0.000 | ±0.000 | ±0.000 | ±0.000 | ±0.000 | ±0.000 | ±0.000 | ±0.000 | ±0.000 |
| PCMCI | 0.750 | 0.583 | 0.167 | 0.700 | 0.357 | 0.250 | 0.778 | 0.239 | 0.122 | 0.742 | 0.408 | 0.200 |
| | ±0.000 | ±0.000 | ±0.000 | ±0.000 | ±0.000 | ±0.000 | ±0.000 | ±0.000 | ±0.000 | ±0.000 | ±0.000 | ±0.000 |
| TCDF | 0.750 | 0.745 | 0.150 | 0.713 | 0.479 | 0.205 | 0.920 | 0.567 | 0.099 | 0.663 | 0.377 | 0.185 |
| | ±0.312 | ±0.264 | ±0.123 | ±0.047 | ±0.069 | ±0.028 | ±0.015 | ±0.094 | ±0.015 | ±0.102 | ±0.101 | ±0.024 |
| Latent CCM | 0.625 | 0.547 | 0.230 | 0.561 | 0.419 | 0.220 | 0.560 | 0.171 | 0.122 | 0.550 | 0.296 | 0.200 |
| | ±0.153 | ±0.162 | ±0.075 | ±0.203 | ±0.199 | ±0.063 | ±0.057 | ±0.028 | ±0.000 | ±0.139 | ±0.087 | ±0.000 |
| CUTS | 0.569 | 0.594 | 0.222 | 0.490 | 0.374 | 0.222 | 0.798 | 0.343 | 0.119 | 0.622 | 0.348 | 0.190 |
| | ±0.360 | ±0.312 | ±0.144 | ±0.218 | ±0.168 | ±0.036 | ±0.068 | ±0.081 | ±0.006 | ±0.111 | ±0.096 | ±0.022 |
| CUTS+ | 0.554 | 0.519 | 0.262 | 0.557 | 0.433 | 0.200 | 0.749 | 0.390 | 0.100 | 0.753 | 0.447 | 0.180 |
| | ±0.142 | ±0.145 | ±0.089 | ±0.085 | ±0.073 | ±0.024 | ±0.016 | ±0.008 | ±0.000 | ±0.072 | ±0.103 | ±0.026 |
| CSL-HNTS | 0.850 | 0.775 | 0.117 | 0.527 | 0.270 | 0.250 | 0.552 | 0.140 | 0.122 | 0.512 | 0.243 | 0.200 |
| | ±0.219 | ±0.304 | ±0.158 | ±0.060 | ±0.045 | ±0.000 | ±0.055 | ±0.018 | ±0.000 | ±0.150 | ±0.064 | ±0.000 |
| AERCA | 0.542 | 0.536 | 0.250 | 0.548 | 0.344 | 0.240 | 0.858 | 0.432 | 0.110 | 0.552 | 0.335 | 0.185 |
| | ±0.151 | ±0.146 | ±0.091 | ±0.144 | ±0.088 | ±0.021 | ±0.040 | ±0.102 | ±0.011 | ±0.134 | ±0.111 | ±0.024 |
| TopoDistill-TCN | 0.888 | 0.845 | 0.067 | **0.844** | 0.648 | 0.190 | **0.945** | **0.685** | **0.084** | 0.794 | 0.508 | 0.155 |
| | ±0.150 | ±0.203 | ±0.086 | ±0.013 | ±0.051 | ±0.021 | ±0.010 | ±0.079 | ±0.012 | ±0.025 | ±0.051 | ±0.016 |
| TopoDistill-MLP | **0.912** | **0.908** | **0.050** | 0.825 | **0.647** | **0.185** | 0.940 | 0.627 | 0.089 | **0.795** | **0.525** | **0.150** |
| | ±0.167 | ±0.159 | ±0.081 | ±0.018 | ±0.057 | ±0.024 | ±0.013 | ±0.089 | ±0.014 | ±0.030 | ±0.053 | ±0.000 |

This weighted average mimics the simplex projection used in classical CCM, but operates in the learned embedding space $\mathbb{R}^{D_s}$ rather than delay coordinates. For each library size $L_j$, we compute the Pearson correlation $\rho_j$ between predicted values and true values across the test set. This yields a convergence curve $\{\rho_j\}_{j=1}^m$ as a function of $L_j$.

In noisy or finite-sample scenarios, convergence curves may exhibit fluctuations. We consider two aspects: the area under the curve (AUC) and the convergence trend (Appendix D).

Convergence trend captures the monotonic increase characteristic of true causality:

$$\text{Trend} = \underbrace{(\rho_{\text{tail}} - \rho_{\text{head}})}_{\text{Magnitude}} \times \underbrace{|\tau_{\text{Kendall}}(L, \rho)|}_{\text{Consistency}}, \quad (19)$$

where $\rho_{\text{head}}$ and $\rho_{\text{tail}}$ denote the correlation coefficients at the smallest and largest library sizes, respectively. $\tau_{\text{Kendall}}(L, \rho)$ is Kendall's rank correlation coefficient, measuring monotonicity. The final CCM score is computed as:

$$\mathcal{S}_{\text{CCM}} = \text{AUC}(\{\rho_j\}) + \text{Trend}(\{\rho_j\}). \quad (20)$$

The pseudocode for the overall procedure is provided in Appendix E. An extended discussion on the noise robustness of our framework, including its macroscopic denoising mechanisms and performance under temporally correlated noise, is provided in Appendix F.

## 5. Experiments

To evaluate our proposed TopoDistill framework, we compare our method against eight popular baseline approaches on three synthetic datasets and one real-world dataset.

### 5.1. Experimental Setup

**Datasets.** Experiments are conducted on three synthetic datasets from different domains: Double Pendulum (De Brouwer et al., 2020), UUMC (Herman et al., 2025), fMRI (Smith et al., 2011), and one real-world dataset: Causal Rivers (Stein et al., 2025).

**Baseline Methods.** The baseline methods include CCM (Sugihara et al., 2012), PCMCI (Runge et al., 2019), TCDF (Nauta et al., 2019), Latent CCM (De Brouwer et al., 2020), CUTS (Cheng et al., 2023), CUTS+ (Cheng et al., 2024), CSL-HNTS (Chen et al., 2024), AERCA (Han et al., 2025).

**Metrics.** Since CCM-based approaches lack an explicit decision threshold, we report threshold-free performance using AUC-ROC and AUC-PR. We also report SHD under the optimal empirical threshold to estimate the best achievable graph recovery. Additional details are provided in Appendix G.

### 5.2. Causal Discovery Performance

The results of our framework and baseline methods are summarized in Table 1. For the architecture of the delay embedder, we consider two basic backbones: TCN and MLP. Our method achieves the best performance across all evaluation metrics, and the two delay embedder variants deliver comparable results. For CCM and PCMCI, since they do not rely on random initialization, their results are deterministic and exhibit no run-to-run variability.

Compared with conventional CCM, our method achieves a substantial improvement in terms of SHD, and delivers clear

gains in PR on most datasets. Notably, Causal Rivers is a newly introduced and particularly challenging real-world dataset, on which all baseline methods struggle. This may be because the dataset contains considerable noise and exhibits long-range temporal dependencies.

### 5.3. Ablation Study

In the ablation study, we separately remove topological distillation and temporal contrastive learning, and further remove both while using a window reconstruction objective to learn window representations. Results of the ablation study are reported in Table 2.

After removing topological distillation (-Topo), the delay embedder is trained solely under the contrastive learning paradigm. Although it can still learn sequence representations to some extent, these representations are not aligned with the underlying system manifold and cannot effectively exploit multivariate information, leading to degraded causal discovery performance. After removing temporal contrastive learning (-Smooth), the explicit constraint on temporal continuity is eliminated, resulting in a rougher temporal trajectory of the shadow manifold.

When both losses are removed (Rec), we train the delay embedder using a reconstruction objective as a proxy task, which is a common and reasonable practice in time-series representation learning. Although this objective can yield sequence representations, it imposes no constraint on the shadow manifold itself. Consequently, the learned representations may be ill-suited for CCM-based causal discovery. On the more challenging real-world dataset, the reconstruction paradigm causes a substantial performance drop. The complete results of the ablation studies are provided in Appendix G.5.

### 5.4. Analysis on Lorenz Dataset

To evaluate the robustness of our method under noise and further compare it with TDE, we conducted experiments on the Lorenz–96 dataset (Lorenz, 1996) with 10 variables. The dataset has a maximum library length of 720. We add

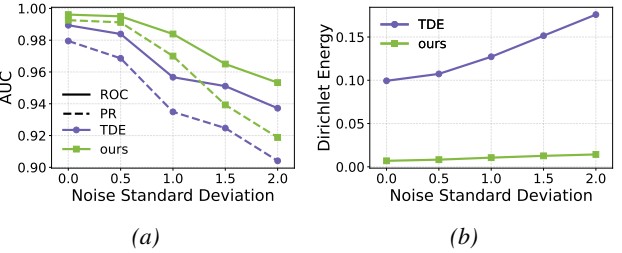

*Figure 3.* On datasets with noise of varying standard deviations, our method was compared with TDE in terms of (a) causal discovery performance and (b) neighborhood Dirichlet energy.

*Table 2.* Ablation study on fMRI and Causal Rivers (CR) datasets.

| Dataset | Metric | Full | -Topo | -Smooth | Rec |
|---------|--------|------|-------|---------|-----|
| fMRI | ROC | **0.949** | 0.938 | 0.937 | 0.934 |
| | PR | **0.744** | 0.596 | 0.642 | 0.551 |
| | SHD | **0.067** | 0.089 | 0.100 | 0.100 |
| CR | ROC | **0.813** | 0.766 | 0.734 | 0.672 |
| | PR | **0.508** | 0.415 | 0.452 | 0.323 |
| | SHD | **0.150** | 0.200 | 0.200 | 0.200 |

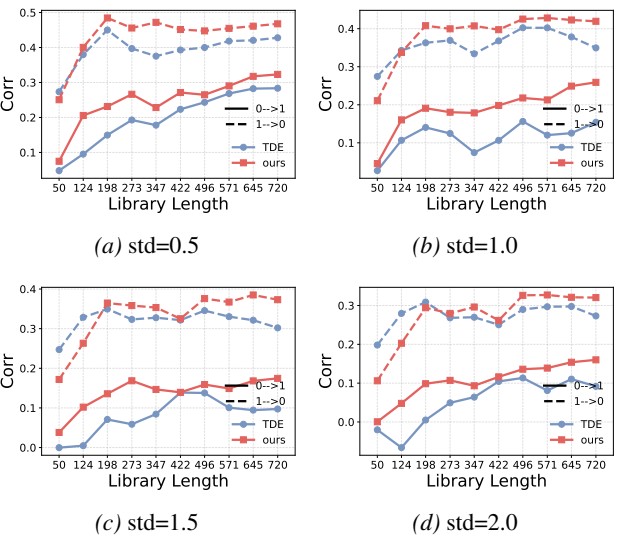

*(a)* std=0.5      *(b)* std=1.0

*(c)* std=1.5      *(d)* std=2.0

*Figure 4.* Cross-mapping skill between a pair of bidirectionally causal variables under noise with varying standard deviations.

Gaussian noise with standard deviations ranging from 0.5 (approximately 11.5%) to 2.0 (approximately 46%). Fig. 3a compares the performance of TopoDistill and TDE. The two methods use an identical CCM pipeline and differ only in how the shadow manifold is constructed. Across different noise levels, our method consistently outperforms TDE, demonstrating strong robustness to noise.

To examine the effect of noise on the shadow manifold, we further measure local neighborhood smoothness by the graph Dirichlet energy (Appendix G.4) of the embeddings on a $k$-NN graph, normalized by a Rayleigh quotient. This normalization makes the metric insensitive to global scaling of the embedding. This value reflects, to some extent, the local smoothness of the shadow manifold: smaller values indicate a smoother manifold. The results are shown in Fig. 3b; even under relatively high noise levels, the shadow manifolds generated by our method remain reasonably smooth.

To more clearly illustrate the advantages of our method on noisy data, we select a pair of variables with a bidirectional causal relationship in the Lorenz–96 system and visualize the convergence curves of cross mapping between them.

Ideally, as the library length increases, the correlation between the observed variable and the cross-mapped estimate of its cause should increase and eventually converge to a plateau. Fig. 4 shows the convergence curves under different noise levels, with the correlation coefficient on the Corr-axis. Under noise, our method exhibits a more distinct plateau, indicating better convergence of the correlation, whereas TDE is comparatively more unstable. At higher noise levels, although the final converged value decreases, it still converges to a higher level than TDE and does so more stably.

To provide a transparent view of the model's computational footprint in practical deployments, a comprehensive analysis of the computational complexity, efficiency, and the performance-efficiency trade-off on this dataset is detailed in Appendix H.

## 6. Conclusion

In this paper, we proposed TopoDistill, a novel topology-guided knowledge distillation framework for robust causal discovery in multivariate time series. To address the geometric degradation of shadow manifolds constructed by standard time-delay embedding under noise, our method leverages a global embedder to capture an implicitly denoised, system-level topological template. By distilling this global neighborhood structure into a univariate delay embedder, and regularizing it with a temporal contrastive objective, TopoDistill yields smooth and dynamically faithful shadow manifolds for convergent cross mapping. Theoretical and experimental analyses confirm that TopoDistill maintains strong robustness against complex measurement noise while preserving computational efficiency, offering a highly effective tool for uncovering causal mechanisms in real-world dynamical systems.

**Limitations.** TopoDistill requires training a neural network and may fail when only limited data are available. Future work could incorporate few-shot learning techniques to mitigate this issue. In addition, currently no automated procedure for selecting an empirical threshold is provided. In practice, this issue can be partially mitigated by inspecting the convergence curves. Future work could explore adaptive selection of empirical thresholds across different datasets.

## Acknowledgments

This work was supported in part by the Xinmiao Project of ZheJiang Provincial Applied Basic Research Program under Grant No. 2026XMHD003, in part by the "Pioneer" and "Leading Goose" R&D Program of Zhejiang No. 2024C01212, and in part by the National Natural Science Foundation of China under Grant No. 62372146.

## Impact Statement

This paper presents work whose goal is to advance the field of Machine Learning. There are many potential societal consequences of our work, none which we feel must be specifically highlighted here.

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

# A. Mathematical Foundations of Global Embedder

## A.1. Koopman Operator Theory and Problem Setup

Consider a deterministic dynamical system $\mathbf{x}_{t+1} = F(\mathbf{x}_t)$ on a compact attractor $\mathcal{A} \subset \mathbb{R}^n$, where $F : \mathcal{A} \to \mathcal{A}$ is smooth. The **Koopman operator** $\mathcal{K}$ acts on scalar observables $g : \mathcal{A} \to \mathbb{R}$ via $(\mathcal{K}g)(\mathbf{x}) = g(F(\mathbf{x}))$. While $F$ may be nonlinear, $\mathcal{K}$ is a linear operator on the infinite-dimensional function space $\mathcal{F}(\mathcal{A})$.

For practical computation, we seek finite-dimensional approximations. A vector of observables $\phi = [\phi_1, \ldots, \phi_{D_g}]^T : \mathcal{A} \to \mathbb{R}^{D_g}$ spans an *approximately Koopman-invariant subspace* if

$$\phi(F(\mathbf{x})) \approx \mathbf{K}\phi(\mathbf{x}), \quad \forall \mathbf{x} \in \mathcal{A}, \tag{21}$$

for some matrix $\mathbf{K} \in \mathbb{R}^{D_g \times D_g}$, implying linearized dynamics in the embedding space.

Our global embedder consists of encoder $f_g : \mathbb{R}^n \to \mathbb{R}^{D_g}$ and decoder $f_{\text{dec}} : \mathbb{R}^{D_g} \to \mathbb{R}^n$, trained via:

$$\min_{\theta_g, \theta_d} \mathcal{L}_{\text{global}} = \mathbb{E}_{\mathbf{x}_t \sim p_{\mathcal{A}}}[\|\mathbf{x}_{t+1} - f_{\text{dec}}(f_g(\mathbf{x}_t))\|^2]. \tag{22}$$

## A.2. Main Theoretical Result

**Proposition A.1** (Dynamical Sufficiency of Global Embeddings). *Suppose the encoder-decoder pair $(f_g, f_{dec})$ achieves prediction error*

$$\mathbb{E}_{\mathbf{x} \sim p_{\mathcal{A}}}[\|F(\mathbf{x}) - f_{dec}(f_g(\mathbf{x}))\|^2] \leq \epsilon. \tag{23}$$

*Then the embedding $\mathbf{z}_t = f_g(\mathbf{x}_t)$ satisfies:*

1. *(**Information Preservation**) $f_g(\mathbf{x})$ retains all dynamically-relevant information: for any predictor $h : \mathbb{R}^n \to \mathbb{R}^n$, there exists $\tilde{h} : \mathbb{R}^{D_g} \to \mathbb{R}^n$ such that*

$$\mathbb{E}[\|F(\mathbf{x}) - \tilde{h}(f_g(\mathbf{x}))\|^2] \leq \mathbb{E}[\|F(\mathbf{x}) - h(\mathbf{x})\|^2] + C\epsilon. \tag{24}$$

2. *(**Koopman Invariance**) There exists $\mathbf{K} \in \mathbb{R}^{D_g \times D_g}$ such that*

$$\mathbb{E}[\|f_g(F(\mathbf{x})) - \mathbf{K}f_g(\mathbf{x})\|^2] \leq C\epsilon, \tag{25}$$

   *i.e., the dynamics are approximately linearized in the embedding space.*

3. *(**Topological Preservation**) The map $f_g : \mathcal{A} \to \mathbb{R}^{D_g}$ is an $\epsilon$-homeomorphism, preserving the topological structure of trajectories up to $\mathcal{O}(\epsilon)$ distortion.*

*Proof.* **Part 1 (Information Sufficiency):** By the tower property of conditional expectation, the optimal predictor based on $f_g(\mathbf{x})$ is $\tilde{h}^*(\mathbf{z}) = \mathbb{E}[F(\mathbf{x})|f_g(\mathbf{x}) = \mathbf{z}]$. For deterministic systems, $F(\mathbf{x})$ is the optimal predictor given $\mathbf{x}$. The prediction loss (22) ensures $f_{\text{dec}}(f_g(\mathbf{x})) \approx F(\mathbf{x})$, which by standard results in information theory implies that $f_g(\mathbf{x})$ is approximately sufficient for predicting $F(\mathbf{x})$.

Formally, decomposing the prediction error:

$$\mathbb{E}[\|F(\mathbf{x}) - \tilde{h}^*(f_g(\mathbf{x}))\|^2] = \mathbb{E}[\|F(\mathbf{x}) - \mathbb{E}[F(\mathbf{x})|f_g(\mathbf{x})]\|^2]. \tag{26}$$

Since $f_{\text{dec}}$ approximates the conditional expectation with error $\leq \epsilon$, the information loss is bounded by $\mathcal{O}(\epsilon)$.

**Part 2 (Koopman Invariance):** Define the optimal linear operator:

$$\mathbf{K}^* = \mathbb{E}[f_g(F(\mathbf{x}))f_g(\mathbf{x})^T] \left(\mathbb{E}[f_g(\mathbf{x})f_g(\mathbf{x})^T]\right)^{-1}. \tag{27}$$

By the training condition, $F(\mathbf{x}) \approx f_{\text{dec}}(f_g(\mathbf{x}))$. Applying $f_g$ and using the Lipschitz continuity of $f_g$ with constant $L_g$:

$$\mathbb{E}[\|f_g(F(\mathbf{x})) - f_g(f_{\text{dec}}(f_g(\mathbf{x})))\|^2] \leq L_g^2 \mathbb{E}[\|F(\mathbf{x}) - f_{\text{dec}}(f_g(\mathbf{x}))\|^2] \leq L_g^2 \epsilon. \tag{28}$$

When $f_g \circ f_{\text{dec}} \approx \text{Id}$ (approximate autoencoder property), we have $f_g(f_{\text{dec}}(\mathbf{z})) \approx \mathbf{z}$. By the optimality of $\mathbf{K}^*$ for linear approximation and the above bound, $\mathbb{E}[\|f_g(F(\mathbf{x})) - \mathbf{K}^* f_g(\mathbf{x})\|^2] \leq C\epsilon$ for $C = L_g^2 + 1$.

**Part 3 (Topological Preservation):** Neural networks with smooth activations are continuous. The prediction accuracy enforces approximate injectivity: if $f_g(\mathbf{x}_1) = f_g(\mathbf{x}_2)$, then by the triangle inequality,

$$\|F(\mathbf{x}_1) - F(\mathbf{x}_2)\| \leq 2\sqrt{\epsilon}. \tag{29}$$

On a compact attractor with bounded dynamics, this implies $\mathbf{x}_1$ and $\mathbf{x}_2$ must have similar dynamical futures. The prediction task thus forces $f_g$ to be approximately injective on dynamically distinct states. Combined with continuity, $f_g$ forms an $\epsilon$-homeomorphism onto its image, preserving topological structure. $\square$

*Remark* A.2. Proposition A.1 shows that the global embedder learns a Koopman-like representation where: (i) encoder components are nonlinear observables $\phi_i(\mathbf{x})$, (ii) their evolution is approximately linear ($\mathbf{z}_{t+1} \approx \mathbf{K}\mathbf{z}_t$), and (iii) the embedding preserves topological structure. This justifies using $\{\mathbf{z}_t^g\}$ as a template for guiding shadow manifold reconstruction.

### A.3. Implications for Shadow Manifold Alignment

The results above directly motivate our topology distillation loss. Since:

1. $\{\mathbf{z}_t^g\}$ faithfully represents the system manifold topology (Prop. A.1, Part 3),

2. The local neighborhood structure in $\{\mathbf{z}_t^g\}$ reflects true dynamical proximity (Prop. A.1, Part 2),

By aligning the pairwise distance distribution of shadow embeddings $\{\mathbf{z}_t^{(i)}\}$ with that of $\{\mathbf{z}_t^g\}$, we effectively guide the single-series encoder to:

- Reconstruct a shadow manifold with geometric fidelity to the true system manifold,

- Suppress noise-induced distortions that would otherwise corrupt distance-based CCM predictions,

- Preserve the causal structure encoded in the dynamical proximity relationships.

This completes the theoretical foundation for our topology-guided embedding framework.

**Why not use raw multivariate snapshots directly?** While $\mathbf{x}_t \in \mathbb{R}^n$ contains complete system information, directly using it as the topological template poses several challenges:

1. *Noise contamination*: Observation noise $\epsilon_t$ in $\mathbf{x}_t = \mathbf{x}_t^* + \epsilon_t$ directly pollutes distance computations, as noise components do not carry predictive information about future states.

2. *Dimension mismatch*: For high-dimensional systems ($n \gg d_A$, where $d_A$ is the attractor dimension), comparing a shadow manifold $\mathcal{M}_i \subset \mathbb{R}^{D_s}$ with $D_s \sim 2d_A + 1$ to the full observation space $\mathbb{R}^n$ is geometrically ill-posed.

3. *Metric distortion*: Euclidean distances in the observation space may not reflect dynamical proximity due to variable scaling, units, and nonlinear couplings.

In contrast, our global embedder $f_g : \mathbb{R}^n \to \mathbb{R}^{D_g}$ addresses these issues by:

- Extracting noise-robust features through the prediction objective, which filters out components uncorrelated with future evolution,

- Compressing to a dimension $D_g \approx d_E$ (embedding dimension) that matches the intrinsic complexity of the attractor,

- Learning a task-relevant metric where distances reflect dynamical similarity rather than raw observation differences.

Mathematically, $f_g$ learns nonlinear Koopman observables that linearize the dynamics, providing an optimal "coordinate system" for topological comparison.

## B. Mathematical Justification for Probabilistic Neighborhood Encoding

In Section 4.3, we use conditional probability distributions to encode local topology on manifolds:

$$P_{\mathcal{M}}(j|i) = \frac{\exp(-\|m_i - m_j\|^2/\sigma^2)}{\sum_{k \neq i} \exp(-\|m_i - m_k\|^2/\sigma^2)} \tag{30}$$

where $m_i$ denotes the coordinate of point $i$ on manifold $\mathcal{M}$, and $\sigma$ is a bandwidth parameter. This appendix provides the theoretical foundation for this formulation.

### B.1. Motivation: From Hard Neighborhoods to Soft Probabilities

In classical topology, a neighborhood is defined as a binary relation:

$$\mathcal{N}_\epsilon(i) = \{j : d(i,j) < \epsilon\} \tag{31}$$

While theoretically rigorous, this hard thresholding is non-differentiable and unsuitable for gradient-based optimization. The probabilistic formulation provides a *soft* alternative where the probability of $j$ being a neighbor of $i$ decays smoothly with distance. This construction has been widely adopted in manifold learning methods such as stochastic neighbor embedding and t-SNE, where it successfully preserves local structure in dimensionality reduction.

### B.2. Theoretical Perspectives

#### B.2.1. HEAT KERNEL INTERPRETATION

On a Riemannian manifold $\mathcal{M}$, the *heat kernel* $p_t(x, y)$ describes heat diffusion and satisfies the heat equation:

$$\frac{\partial}{\partial t} p_t(x, y) = \Delta_{\mathcal{M}} p_t(x, y) \tag{32}$$

where $\Delta_{\mathcal{M}}$ is the Laplace-Beltrami operator. For small diffusion time $t$, the heat kernel admits the asymptotic expansion:

$$p_t(x, y) \approx \frac{1}{(4\pi t)^{d/2}} \exp\left(-\frac{d_{\mathcal{M}}^2(x, y)}{4t}\right) \tag{33}$$

where $d_{\mathcal{M}}(x, y)$ is the geodesic distance on $\mathcal{M}$.

Our conditional probability can be viewed as a normalized, discretized heat kernel with $\sigma^2 \leftrightarrow 2t$. This connection is significant because:

- The heat kernel is intrinsic to the manifold geometry and does not depend on embedding coordinates.

- For small $t$ (or $\sigma$), the heat kernel primarily encodes *local geodesic distances*, making it a natural descriptor of local topology.

- Aligning heat kernels across two manifolds ensures that local diffusion processes—and hence intrinsic geometries—are preserved.

#### B.2.2. RANDOM WALK AND DIFFUSION DYNAMICS

The conditional probability $P(j|i)$ can be interpreted as a one-step transition probability in a random walk on the manifold. Define the transition matrix:

$$P_{ij} = P(j|i) = \frac{w_{ij}}{\sum_k w_{ik}}, \quad w_{ij} = \exp(-\|m_i - m_j\|^2/\sigma^2) \tag{34}$$

The theory of diffusion maps establishes that the eigendecomposition of $P$ yields coordinates that preserve *diffusion distances*:

$$D_t^2(i, j) = \sum_k (p_t(i, k) - p_t(j, k))^2 \tag{35}$$

where $p_t(i, k)$ is the $t$-step transition probability. Crucially, diffusion distance approximates geodesic distance on the manifold when samples are dense. Therefore, aligning transition probabilities across two spaces ensures their random walk dynamics—and thus local connectivity—are consistent.

B.2.3. Graph Laplacian and Spectral Convergence

Define the affinity matrix $W$ with entries $W_{ij} = \exp(-d_{ij}^2/\sigma^2)$, and the degree matrix $D$ with $D_{ii} = \sum_j W_{ij}$. The normalized graph Laplacian is:

$$L = I - D^{-1}W \tag{36}$$

Spectral graph theory establishes that as sample density $n \to \infty$ and bandwidth $\sigma \to 0$ (at appropriate relative rates), the graph Laplacian converges to the Laplace-Beltrami operator:

$$L \xrightarrow[n \to \infty]{} \Delta_{\mathcal{M}} \tag{37}$$

Since the spectrum of $\Delta_{\mathcal{M}}$ encodes topological invariants (e.g., the number of connected components, Betti numbers), aligning the transition probabilities $P(j|i)$ effectively aligns the spectra of the corresponding graph Laplacians, thereby preserving topological structure.

## B.3. Limitations and Practical Considerations

**Not a Sufficient Condition for Diffeomorphism.** We emphasize that aligning $P_{\mathcal{M}_1}(j|i) \approx P_{\mathcal{M}_2}(j|i)$ is *not* a sufficient condition for the manifolds $\mathcal{M}_1$ and $\mathcal{M}_2$ to be diffeomorphic. Counter-examples exist where locally similar neighborhoods do not imply global topological equivalence (e.g., a torus and a sphere can have similar local metrics but different global topology).

**Why It Works in Practice.** Despite this limitation, our approach is effective because:

1. **Combined Constraints:** We do not rely solely on neighborhood alignment. The contrastive loss (Section **??**) explicitly enforces global unfolding and temporal continuity, providing complementary constraints that approximate diffeomorphism.

2. **Statistical Validity:** For CCM, which operates via $k$-nearest neighbors, preserving *most* local neighborhoods in a statistical sense suffices. We do not require pointwise exactness.

3. **Empirical Success:** The probabilistic neighborhood encoding has been validated in numerous manifold learning applications, demonstrating robust topology preservation in practice.

**Role of Temperature $\sigma$.** The bandwidth parameter $\sigma$ (or equivalently, temperature $T$ in our implementation) controls the locality-globality trade-off:

- Small $\sigma$: Only immediate neighbors have non-negligible probability, emphasizing fine-grained local structure.

- Large $\sigma$: Probabilities spread across distant points, capturing broader geometric relationships.

In practice, we set $\sigma$ to balance these effects, often tuning it as a hyperparameter or adapting it per-point.

## B.4. Summary

The conditional probability formulation $P(j|i) \propto \exp(-d_{ij}^2/\sigma^2)$ is grounded in:

- **Heat kernel theory**: encoding local geodesic structure via diffusion processes.

- **Random walk theory**: preserving intrinsic connectivity through transition probabilities.

- **Spectral graph theory**: aligning topological invariants via graph Laplacian spectra.

While not strictly equivalent to enforcing diffeomorphism, this approach provides a differentiable, computationally tractable, and empirically validated proxy for topology preservation. When combined with our contrastive learning objective, it enables effective manifold alignment for CCM-based causal discovery.

## C. Mathematical Justification for Topology Distilling

### C.1. Approximate Homeomorphism Under Noise

Consider a dynamical system on a compact $d$-dimensional attractor $\mathcal{A} \subset \mathbb{R}^n$ with noisy observations $\tilde{\mathbf{x}}_t = \mathbf{x}_t + \boldsymbol{\epsilon}_t$, where $\boldsymbol{\epsilon}_t$ is bounded noise ($\|\boldsymbol{\epsilon}_t\| \leq \delta$) independent of system dynamics. Define $T_{\mathbf{x}}\mathcal{A}$ as the tangent space (dynamical directions) and $N_{\mathbf{x}}\mathcal{A} = (T_{\mathbf{x}}\mathcal{A})^{\perp}$ as the normal space (noise directions).

**Theorem C.1** (Noise-Robust Homeomorphism). *Suppose the neural embedder $f_s : \mathbb{R}^w \to \mathbb{R}^{D_s}$ satisfies:*

1. ***Tangential preservation:** $\|df_s(\mathbf{v})\| \geq c\|\mathbf{v}\|$ for $\mathbf{v} \in T_{\mathbf{x}}\mathcal{A}$, some $c > 0$,*

2. ***Normal compression:** $\|df_s(\mathbf{n})\| \leq C\delta\|\mathbf{n}\|$ for $\mathbf{n} \in N_{\mathbf{x}}\mathcal{A}$,*

3. ***Lipschitz continuity:** $\|f_s(\mathbf{x}_1) - f_s(\mathbf{x}_2)\| \leq L\|\mathbf{x}_1 - \mathbf{x}_2\|$.*

*Then $f_s|_{\mathcal{A}}$ is an $\mathcal{O}(\delta)$-homeomorphism with:*

- ***Injectivity:** $\|f_s(\mathbf{x}_1) - f_s(\mathbf{x}_2)\| \geq c\|\mathbf{x}_1 - \mathbf{x}_2\| - 2C\delta$ for $\mathbf{x}_1, \mathbf{x}_2 \in \mathcal{A}$,*

- ***Noise insensitivity:** $\|f_s(\tilde{\mathbf{x}}) - f_s(\mathbf{x})\| = \mathcal{O}(\delta)$ for $\tilde{\mathbf{x}} = \mathbf{x} + \boldsymbol{\epsilon}$.*

*Proof.* Decompose displacements into tangential and normal components: $\mathbf{x}_2 - \mathbf{x}_1 = \mathbf{v}_{\parallel} + \mathbf{v}_{\perp}$. For points on $\mathcal{A}$, $\|\mathbf{v}_{\perp}\| = \mathcal{O}(\|\mathbf{v}_{\parallel}\|^2)$ (manifold structure). Applying Conditions 1-2:

$$\|f_s(\mathbf{x}_2) - f_s(\mathbf{x}_1)\| \geq \|df_s(\mathbf{v}_{\parallel})\| - \|df_s(\mathbf{v}_{\perp})\| \geq c\|\mathbf{v}_{\parallel}\| - C\delta\|\mathbf{v}_{\perp}\| \geq c\rho - 2C\delta, \tag{38}$$

ensuring injectivity when $c\rho > 2C\delta$. For noise robustness, since $\boldsymbol{\epsilon}$ is predominantly normal to $\mathcal{A}$, Condition 2 yields $\|f_s(\tilde{\mathbf{x}}) - f_s(\mathbf{x})\| \leq L\delta + C\delta^2 = \mathcal{O}(\delta)$. $\square$

**Corollary C.2** (Neighborhood Preservation). *Under Theorem C.1, if $\mathbf{x}_i, \mathbf{x}_j$ are $k$-nearest neighbors on $\mathcal{A}$, then $f_s(\mathbf{x}_i), f_s(\mathbf{x}_j)$ are approximate $k$-nearest neighbors in $\mathbb{R}^{D_s}$ up to $\mathcal{O}(\delta)$ error. This ensures CCM's distance-based prediction reflects true dynamical proximity.*

**Connection to topology distillation.** The topology alignment loss $\mathcal{L}_{\text{topo}} = \text{KL}(P^g \| Q^{(i)})$ implicitly enforces Theorem C.1's conditions:

- Matching neighborhood distributions $P^g$ (from denoised global embeddings) teaches $f_s$ to preserve tangential structure while ignoring normal noise components.

- The soft constraints avoid explicit Jacobian regularization, letting the network learn anisotropic behavior naturally.

### C.2. Effectiveness of Global Topology Distillation

**Proposition C.3** (Information Advantage of Multivariate Observations). *Let $I_{global} = I(\mathbf{x}_t; \mathbf{s}_t)$ and $I_{local} = I(\mathbf{w}_t^{(i)}; \mathbf{s}_t)$ denote mutual information between the latent state $\mathbf{s}_t$ and multivariate observations $\mathbf{x}_t$ vs. single-variable windows $\mathbf{w}_t^{(i)}$. Under generic observation functions and independent noise, $I_{global} \geq I_{local}$, with strict inequality when $n \geq d$ (attractor dimension).*

*Proof.* By the data processing inequality, $I(\mathbf{w}_t^{(i)}; \mathbf{s}_t) \leq I(\{\mathbf{x}_{\tau}\}_{\tau \leq t}; \mathbf{s}_t)$. A single-variable window captures only a 1D projection of $\mathbf{s}_t$, while multivariate snapshots observe $n$ independent projections simultaneously. For $n \geq d$, these projections span the full attractor, yielding $I_{\text{global}} \sim \frac{1}{2}\log\det(\mathbf{I} + \boldsymbol{\Sigma}_{\mathbf{s}}/\sigma^2) \gg I_{\text{local}} \sim \frac{1}{2}\log(1 + \sigma_{\text{proj}}^2/\sigma^2)$. $\square$

**Proposition C.4** (Topology Transfer via Distillation). *Let $P^{true}, P^g, Q^{(i)}$ denote neighborhood distributions for the true attractor, global embedding, and shadow manifold. If the global encoder achieves $\text{KL}(P^{true} \| P^g) \leq \epsilon_g$ and topology distillation achieves $\text{KL}(P^g \| Q^{(i)}) \leq \epsilon_s$, then:*

$$KL(P^{true} \| Q^{(i)}) \leq \epsilon_g + \epsilon_s. \tag{39}$$

*Proof.* By the chain rule for KL divergence: $\text{KL}(P^{\text{true}} \| Q^{(i)}) = \text{KL}(P^{\text{true}} \| P^g) + \text{KL}(P^g \| Q^{(i)}) \leq \epsilon_g + \epsilon_s$. $\square$

**Implication.** Proposition C.4 shows that shadow manifolds inherit the topological fidelity of global embeddings through distillation. Since $\epsilon_g$ is small (Prop. C.3) and $\epsilon_s$ is minimized during training, $f_s$ indirectly aligns with the true attractor topology $P^{\text{true}}$, despite never observing the latent state directly.

$\square$

## D. Method Details

### D.1. Delay Embedder

The delay embedder is designed to be shared across all sequences. Under Takens' theorem, the homeomorphic property of TDE is independent of the specific observation variable. That is, within the same coupled system, delay embeddings constructed from different variables reconstruct the same underlying attractor from different viewpoints. Accordingly, we employ a shared delay embedder to learn a generic mapping from a temporal window to the corresponding shadow manifold, projecting all sequences into a common latent space in a manner consistent with the universality of Takens' theorem.

### D.2. CCM Scoring

To test whether variable $X$ drives $Y$, CCM constructs an embedded manifold from $Y$'s time series as the library and excludes the target time point via cross-validation. For each time point $t$ of $X$, it searches for the $k_{nn}$ nearest neighbors of $y_t$ in $Y$'s manifold embedding and obtains the corresponding set of time indices. Using these indices, CCM directly retrieves the observations from $X$'s raw time series and predicts $x_t$ via a distance-weighted average. If the correlation between the predicted and true values increases as the library size grows, this provides evidence supporting a causal influence $X \to Y$.

In noisy or finite-sample scenarios, convergence curves may exhibit fluctuations. We consider two aspects: the area under the curve (AUC) and the convergence trend. AUC measures the overall prediction skill by integrating the convergence curve:

$$AUC = \frac{1}{L_{max} - L_{min}} \int_{L_{min}}^{L_{max}} \max(0, \rho(L)) dL. \tag{40}$$

Convergence trend captures the monotonic increase characteristic of true causality:

$$Trend = \frac{(\rho_{tail} - \rho_{head})}{Magnitude} \times \frac{|\tau_{Kendall}(L, \rho)|}{Consistency}, \tag{41}$$

where $\rho_{head}$ and $\rho_{tail}$ denote the correlation coefficients at the smallest and largest library sizes, respectively. $\tau_{Kendall}(L, \rho)$ is Kendall's rank correlation coefficient, measuring monotonicity. The final CCM score is computed as $\mathcal{S}_{CCM} = AUC(\{\rho_j\}) + Trend(\{\rho_j\})$.

### D.3. Stability of the Smoothness-Based Early Stopping Criterion

The unsupervised early-stopping criterion based on manifold smoothness ($S_m$) is designed to prevent the delay embedder from overfitting to noise. We guarantee against information collapse—where the manifold might collapse into a degenerate form—through both theoretical constraints and empirical evidence.

**Theoretical Guarantee.** The early-stopping metric $S_m$ is a passive monitor rather than an optimization objective. The actual geometric constraint is the temporal contrastive loss ($\mathcal{L}_{smooth}$), whose InfoNCE negative-sampling denominator actively pushes temporally distant states apart. This mathematically prevents the embeddings from collapsing into a degenerate point, ensuring the chaotic dynamics of the attractor are preserved.

**Empirical Evidence.** Analysis on the Lorenz-96 dataset reveals that $S_m$ follows a characteristic V-shaped curve during training. Initially, $S_m$ decreases as the network learns the deterministic manifold and smooths out high-frequency noise. Subsequently, $S_m$ rebounds as the univariate student begins to overfit local noise to minimize the topology loss ($\mathcal{L}_{topo}$), which reintroduces trajectory jitter. Therefore, early stopping at the $S_m$ inflection point acts as an optimal cutoff to mitigate noise overfitting without losing structural information.

# E. Pseudocode of TopoDistill Framework

---

**Algorithm 1** TopoDistill Framework

---

**Require:** Multivariate time series $\{x_1, x_2, \ldots, x_T\}$ where $x_t \in \mathbb{R}^N$
**Require:** Hyperparameters: $D_g, D_s, \lambda, \Delta t_{\text{pos}}, \Delta t_{\text{neg}}$, patience threshold
**Ensure:** Causal adjacency matrix $\mathbf{A} \in \mathbb{R}^{N \times N}$
  1: **Stage 1: Train Global Embedder**
  2: Initialize global encoder $f_g : \mathbb{R}^N \to \mathbb{R}^{D_g}$ and decoder $f_{\text{dec}} : \mathbb{R}^{D_g} \to \mathbb{R}^N$
  3: **while** not converged **do**
  4:      Compute global embeddings: $z_t^g = f_g(x_t)$ for $t = 1, \ldots, T$
  5:      Compute prediction loss: $\mathcal{L}_{\text{global}} = \mathbb{E}_t[\|x_{t+1} - f_{\text{dec}}(z_t^g)\|^2]$
  6:      Update $f_g$ and $f_{\text{dec}}$ to minimize $\mathcal{L}_{\text{global}}$
  7: **end while**
  8: Freeze $f_g$
  9: **Stage 2: Train Delay Embedder**
10: Initialize delay embedder $f_s : \mathbb{R}^\tau \to \mathbb{R}^{D_s}$ (shared across all variables)
11: Compute teacher distribution $p_{t'|t}^g$
12: **while** not early-stopped **do**
13:      **for** each variable $i = 1, \ldots, N$ **do**
14:          Extract windows: $w_t^{(i)} = [x_{t-\tau}^{(i)}, \ldots, x_{t-1}^{(i)}, x_t^{(i)}]$
15:          Compute shadow embeddings: $z_t^{(i)} = f_s(w_t^{(i)})$ for all $t$
16:          Compute student distribution $q_{t'|t}^{(i)}$
17:          Compute topology loss $\mathcal{L}_{\text{topo}}^{(i)}$
18:          Compute temporal contrastive loss $\mathcal{L}_{\text{smooth}}^{(i)}$
19:      **end for**
20:      Total loss: $\mathcal{L}_s = \lambda \sum_{i=1}^N \mathcal{L}_{\text{topo}}^{(i)} + (1 - \lambda) \sum_{i=1}^N \mathcal{L}_{\text{smooth}}^{(i)}$
21:      Update $f_s$ to minimize $\mathcal{L}_s$
22:      Compute smoothness indicator: $S_m = \sum_i \sum_t \|z_{t+1}^{(i)} - z_t^{(i)}\|^2$
23:      **if** $S_m$ increases beyond patience threshold **then**
24:          **break** {Early stopping}
25:      **end if**
26: **end while**
27: **Causal Discovery via CCM**
28: **for** each pair of variables $(i, j)$ **do**
29:      Construct shadow manifold from variable $j$: $\mathcal{M}_j = \{z_t^{(j)}\}_{t=1}^T$
30:      **for** each library size $L_\ell \in \{L_1, \ldots, L_m\}$ **do**
31:          Perform cross-mapping from $\mathcal{M}_j$ to predict variable $i$
32:          Compute correlation $\rho_\ell$ between predictions and true values
33:      **end for**
34:      Compute CCM score: $S_{\text{CCM}}(i, j) = \text{AUC}(\{\rho_\ell\}) + \text{Trend}(\{\rho_\ell\})$
35: **end for**
36: Construct adjacency matrix $\mathbf{A}$ from CCM scores
     **return A**

---

# F. Extended Discussion on Noise Robustness

## F.1. Macroscopic Mechanism of Topology Preservation and Denoising

While Appendix C provides a rigorous mathematical justification for our framework, it is also instructive to understand the macroscopic mechanisms by which our training objectives implicitly enforce the desired topological properties without requiring computationally expensive explicit Jacobian regularization.

**Specific Topological Properties Preserved.** The preserved "topology" in our framework refers to the system's local neighborhood structure under diffeomorphism. The core objective is to maintain the relative ordering of points within a local neighborhood so that it reflects true dynamical proximity, rather than spurious, noise-induced proximity.

**Implicit Enforcement of Properties.** The training objectives enforce these properties through two primary mechanisms:

- **Normal Compression ($\mathcal{L}_{global}$ and $\mathcal{L}_{topo}$):** The global embedder's one-step-ahead prediction loss ($\mathcal{L}_{global}$) naturally filters out unpredictable measurement noise, creating an implicitly denoised topological template. By minimizing the KL-divergence ($\mathcal{L}_{topo}$) against this teacher distribution, the delay embedder is heavily penalized if its representations expand in normal (noise) directions. This effectively enforces normal compression without the need for explicit structural regularization.

- **Tangential Preservation ($\mathcal{L}_{smooth}$):** Our temporal contrastive loss explicitly pushes temporally distant negative samples apart. This negative sampling prevents representation collapse and stretches the temporal trajectory, naturally maintaining the required positive lower bound for the tangential derivative to preserve the underlying dynamical signal.

**Improvement to CCM.** Convergent Cross Mapping (CCM) relies fundamentally on Euclidean $k$-nearest neighbors. TopoDistill ensures that Euclidean proximity in the embedding space strictly equates to true dynamical similarity, thereby eliminating the interference of spurious, noise-induced neighbors and systematically improving causal discovery.

### F.2. Impact of Temporally Correlated Noise (Red Noise)

The theoretical analysis in Section 4.1 assumes that the measurement noise is independent of future states. However, in real-world dynamical systems, noise often exhibits temporal correlation, commonly known as red noise. If the noise itself follows a predictable pattern, there is a risk that the global teacher might inadvertently model the noise components, thereby transferring noise-induced correlations to the student embedder.

To evaluate the robustness of TopoDistill's implicit denoising mechanism under such conditions, we conducted an experiment using the 10-variable Lorenz-96 system corrupted with $40\%$ AR(1) noise, defined as $n_t = \rho n_{t-1} + \epsilon_t$. We varied the autocorrelation coefficient $\rho$ and observed the impact on causal discovery performance. The results are summarized in Table 3.

*Table 3.* Performance of TopoDistill on the Lorenz-96 dataset with $40\%$ AR(1) noise under varying autocorrelation coefficients ($\rho$).

| Autocorrelation ($\rho$) | AUC-ROC | AUC-PR |
|---|---|---|
| $\rho = 0.0$ (White Noise) | 0.969 | 0.934 |
| $\rho = 0.2$ | 0.974 | 0.951 |
| $\rho = 0.4$ (Peak Performance) | 0.988 | 0.980 |
| $\rho = 0.6$ | 0.988 | 0.975 |
| $\rho = 0.8$ (Extreme Red Noise) | 0.980 | 0.946 |

The results reveal an interesting inverted-U performance curve, illustrating the interplay between noise spectral properties and TopoDistill's inductive biases:

- **Benefit of Moderate Correlation ($\rho \in [0.2, 0.6]$):** Moderate AR(1) noise acts as a low-pass filter. Unlike erratic white noise, this temporally "smoother" perturbation aligns better with our $\mathcal{L}_{smooth}$ (tangential preservation) prior. It makes decoupling the noise from the continuous deterministic manifold structurally easier for the network.

- **Risk of Extreme Correlation ($\rho \geq 0.8$):** As hypothesized, highly correlated noise forms a persistent random walk, introducing a spurious dynamic. The global teacher models this dynamic, which then competes for the student's capacity bottleneck ($\sim 6k$ parameters), causing a slight performance dip compared to peak levels.

Importantly, even under extreme red noise conditions ($\rho = 0.8$), TopoDistill outperforms the pure white noise baseline. This confirms that while highly correlated noise introduces modeling challenges, our topological constraints successfully force the student to prioritize the dominant deterministic causal manifold over stochastic fluctuations.

# G. Experiments Details

## G.1. Datasets

**Double Pendulum.** The double pendulum is a classic nonlinear dynamical system that exhibits rich chaotic behavior. The system consists of two point masses $m_1$ and $m_2$ connected to a pivot point and to each other by massless rods of length $\ell_1$ and $\ell_2$, respectively. The state of the system is characterized by the angles $\theta_1$ and $\theta_2$ of the two rods with respect to the vertical, along with their conjugate angular momenta $p_1$ and $p_2$. Each trajectory is therefore a collection of 4-dimensional vector observations: $x_t = [\theta_1, \theta_2, p_1, p_2]^\top \in \mathbb{R}^4$.

To introduce controlled causal dependencies between two pendulum systems $X$ and $Y$, we incorporate a non-physical asymmetric coupling term into the dynamical equations. Specifically, the momentum update for the first angle in system $Y$ is modified as:

$$\dot{p}_1^Y = -\frac{\partial H^Y}{\partial \theta_1^Y} - 2 \cdot c_{X,Y}(\theta_1^Y - \theta_1^X),$$

where $c_{X,Y}$ is a coupling parameter and $H^Y$ denotes the Hamiltonian of system $Y$. The additional term corresponds to a quadratic potential that creates an attractive force on system $Y$ from system $X$. The causal structure is determined by the coupling coefficients:

- $X$ causes $Y$ if and only if $c_{X,Y} \neq 0$,

- $Y$ causes $X$ if and only if $c_{Y,X} \neq 0$,

- $X$ and $Y$ are causally independent if $c_{X,Y} = c_{Y,X} = 0$.

By varying the coupling parameters $(c_{X,Y}, c_{Y,X})$, we can generate datasets with different causal configurations, including unidirectional causation, bidirectional causation, and no causation. This provides a controlled benchmark for evaluating causal discovery methods on deterministic chaotic systems.

**UUMC.** Standard synthetic SCM generation methods often introduce unintended artifacts such as *varsortability* (sample variance correlates with topological order) and *R²-sortability* (coefficient of determination correlates with causal ordering), which can artificially simplify causal discovery tasks.

We utilize the UUMC (Unitless Unrestricted Markov-Consistent) synthetic dataset, which addresses these issues through a principled parameter sampling scheme satisfying three properties: (1) **unitlessness** - invariance to measurement scale changes, (2) **Markov consistency** - graph structure does not influence local parameter magnitudes, and (3) **unrestrictedness** - no artificial constraints on relative parameter sizes.

Each variable follows a linear additive model:

$$x_i = \sum_{j \in \mathrm{Pa}(i)} \beta_{ji} x_j + \epsilon_i, \quad \epsilon_i \sim \mathcal{N}(0, \sigma_i^2),$$

where $\mathrm{Pa}(i)$ denotes the parent set of variable $i$. Unlike standard approaches that sample coefficients from fixed intervals, UUMC employs a scale-free sampling scheme that eliminates topological information leakage through parameter magnitudes, providing a more realistic and unbiased evaluation benchmark.

**fMRI.** This study utilizes simulated fMRI BOLD time-series data generated using the Dynamic Causal Modelling (DCM) forward model with the nonlinear Balloon Model for hemodynamic simulation. Each node receives binarized external input (activation or rest states) generated via a Poisson process, with neural signals propagating through the network with time delays of approximately 50 milliseconds. The hemodynamic response function (HRF) delay varies randomly across nodes (standard deviation 0.5 seconds), and thermal noise (0.1%-1% of mean signal) is added to the BOLD data.

The dataset includes multiple network topologies ranging from 5 to 50 nodes, with small-world structures. It comprises 28 simulation scenarios with varying parameters including network size (5-50 nodes), session duration (2.5-250 minutes), repetition time (0.25-3 seconds), measurement noise levels, HRF delay variability, neural delays (50-100 milliseconds), and connection strengths (mean 0.4-0.9). To evaluate method robustness, the dataset incorporates various confounding factors such as shared input, global additive confounds, inaccurate ROI definitions, backward connections, cyclic causality, non-stationary connections, and single strong input scenarios.

Each simulation scenario contains 50 independent realizations representing individual subjects, with most simulations consisting of 10-minute fMRI scans (200 time points at TR=3 seconds).

**Causal Rivers.** CausalRivers is a large-scale real-world benchmark for time series causal discovery, featuring river discharge data from over 1,160 measurement stations across eastern Germany (666 stations) and Bavaria (494 stations). The dataset spans from 2019 to 2023 with a 15-minute temporal resolution, providing extensive observational data that captures complex hydrological dynamics.

The benchmark includes two comprehensive causal ground truth graphs constructed from multiple information sources including geographical data and remote sensing. With over 1,000 nodes in the full graphs, direct causal discovery is computationally prohibitive. Instead, CausalRivers provides sampling strategies to generate thousands of subgraphs with flexible node counts and diverse structural properties, including single-sink configurations, root causes, hidden confounders, and connected components.

The dataset naturally exhibits several challenging characteristics relevant to real-world causal discovery: high dimensionality, nonlinearity, non-stationarity, seasonal patterns, hidden confounding (through weather effects), and misalignment between causal time lags and sampling rates. An additional subset featuring a recent extreme precipitation event is included to evaluate robustness under distributional shifts. These properties make CausalRivers a realistic testbed for assessing the practical applicability of causal discovery methods on in-the-wild time series data.

### G.2. Baseline Methods

**Convergent Cross Mapping (CCM).** CCM is a nonlinear causal discovery method based on Takens' theorem, particularly effective for deterministic chaotic systems. It constructs shadow manifolds via time-delay embedding: $m_t^{(x)} = [x_{t-(d-1)\tau}, \ldots, x_{t-\tau}, x_t]^\top$, where $d$ is the embedding dimension and $\tau$ is the time delay. To test if $X$ causes $Y$, CCM performs cross-mapping from $\mathcal{M}_Y$ to predict $X$ using $k$-nearest neighbors. Causality $X \to Y$ is established if the prediction accuracy increases and converges as the library size grows. CCM's performance depends critically on manifold reconstruction quality and is sensitive to observation noise.

**PCMCI.** PCMCI (Peter and Clark Momentary Conditional Independence) is a constraint-based method for time series causal discovery. It operates in two stages: the PC stage identifies potential causal parents by testing pairwise conditional independence at different time lags, and the MCI stage refines the graph by testing each link given all other identified parents. PCMCI uses conditional independence tests such as partial correlation or nonparametric tests, efficiently mitigating spurious correlations in high-dimensional settings while discovering both lagged and contemporaneous causal relationships.

**Temporal Causal Discovery Framework (TCDF).** TCDF is a deep learning-based method for temporal causal discovery that uses attention mechanisms in convolutional neural networks (CNNs) to identify causal relationships and time delays in multivariate time series. The framework trains multiple CNNs, where each network predicts one target time series based on the past values of all observed time series.

During training, attention mechanisms learn to weight the importance of different input time series when making predictions. After training, TCDF interprets these learned attention weights to identify potential causal relationships. To validate discovered relationships, TCDF applies a causal validation step through intervention: it perturbs candidate cause variables and tests whether they affect the predicted target series. Only validated relationships are included in the final temporal causal graph.

TCDF additionally discovers time delays between causes and effects by analyzing the network's internal convolutional filters across different time lags. This allows the method to capture both the existence and temporal structure of causal relationships. By leveraging deep learning's representational power, TCDF is designed to handle complex nonlinear dynamics and exhibit robustness to observation noise.

**LatentCCM.** LatentCCM extends the CCM framework to handle short and sporadically observed time series by performing convergent cross mapping in a learned latent space rather than using delay embeddings. Traditional CCM relies on time-delay embedding and is sensitive to missing values, requiring long uninterrupted observations. This limits its applicability in practical scenarios with short or irregular sampling.

LatentCCM addresses this limitation by learning continuous-time latent representations of the system's state space using GRU-ODE-Bayes, a filtering method that extends Neural ODEs. The model jointly processes all segments of sporadic time series to learn the underlying dynamics. Crucially, the filtering nature ensures no future information leaks into the past,

preserving temporal causality. After learning latent representations for each time series, LatentCCM tests for the existence of continuous mappings between these representations to infer causal direction. This approach eliminates the need for delay embeddings and enables causal discovery from very few observations where traditional CCM would fail.

**CUTS** CUTS is a framework designed for causal discovery from irregular time series data with missing entries. Unlike methods that sequentially handle data imputation and causal inference, CUTS performs these tasks in a mutually boosting manner through an iterative process.

The algorithm alternates between two stages: (a) a latent data prediction stage that uses a Delayed Supervision Graph Neural Network (DSGNN) to impute missing entries based on a current causal graph estimate, and (b) a causal graph fitting stage that infers causal relationships from the completed data using an extended nonlinear Granger causality scheme under sparse constraints. The key insight is that a well-designed neural network can effectively fill missing values given a plausible causal graph, which in turn improves subsequent causal discovery. Through this iterative refinement, CUTS progressively enhances both the quality of imputed data and the accuracy of the discovered causal structure, making it suitable for real-world applications with incomplete observations.

**CUTS+.** CUTS+ extends the CUTS framework to enable scalable causal discovery from high-dimensional irregular time series. While CUTS performs iterative data imputation and causal graph learning, its scalability is limited by component-wise LSTMs and MLPs with redundant parameters, making it difficult to handle datasets with dozens or hundreds of variables.

CUTS+ addresses these limitations through two key techniques: (a) Coarse-to-Fine Discovery (C2FD), which facilitates scalable causal graph optimization by hierarchically refining the search space rather than directly learning large causal graphs, and (b) Message-Passing Graph Neural Network (MPGNN), which replaces the redundant component-wise architecture with an efficient message-passing mechanism for data prediction. Like CUTS, CUTS+ alternates between data imputation and causal discovery stages, but with significantly improved computational efficiency and performance on high-dimensional datasets. This makes CUTS+ applicable to real-world scenarios involving large-scale time series such as gene regulatory networks or environmental monitoring systems.

**CSL-HNTS** CSL-HNTS is a continuous optimization-based method for causal discovery from high-dimensional non-stationary time series. Traditional continuous optimization approaches convert the combinatorial DAG search problem into continuous optimization by parameterizing the adjacency matrix through neural networks and enforcing acyclicity via DAG regularizers. However, these methods face limitations when handling non-stationary time series data.

CSL-HNTS addresses three key challenges: First, it employs recurrent neural network architectures capable of capturing temporal dependencies and non-stationary dynamics in time series, rather than relying on simple feedforward models. Second, it improves the guarantee of acyclicity in the learned causal graph through enhanced DAG constraints or sampling methods that directly operate in the DAG space with reduced search complexity. Third, it introduces an automatic threshold determination mechanism to convert weighted adjacency matrices into Boolean causal graphs, eliminating the need for manually tuned thresholds or time-consuming postprocessing steps. This makes CSL-HNTS suitable for discovering causal structures in complex high-dimensional temporal data with non-stationary characteristics.

**AERCA.** AERCA is an autoencoder-based framework for Granger causal discovery in multivariate time series that explicitly models the distributions of exogenous variables. Unlike conventional causal discovery methods that focus solely on identifying causal structures among endogenous variables, AERCA treats the data generation process within a structural causal model framework where both causal relationships and exogenous variable distributions are modeled.

The framework consists of two components: an encoder that performs abductive reasoning to derive the exogenous variable for each time series, and a decoder that learns deductive reasoning to reconstruct observed data from the exogenous variables. A core assumption is that exogenous variables are mutually independent, which is enforced through specific independence constraints during training. The decoder is theoretically shown to only require exogenous variables and observed time series from a fixed window prior to time $t$ for prediction. By explicitly separating exogenous and endogenous factors, AERCA provides a foundation for identifying Granger causal relationships that accounts for the influence of external driving forces on the system dynamics.

## G.3. Implementation Details

Our method requires training a neural network; therefore, we manually split the data into training and test sets and train the network in a self-supervised manner on the training set. The total amount of data used for training and testing is the same as that used by the other methods. Since CCM-based approaches do not provide an explicit threshold, we evaluate performance using two threshold-free metrics: the Area Under the Curve of the Receiver Operating Characteristic (AUC-ROC) and the Precision-Recall Curve (AUC-PR). In addition, we report the Structural Hamming Distance (SHD) under the optimal threshold to assess the potential performance achievable with the best empirical threshold.

**Structural Hamming Distance (SHD).** Structural Hamming Distance measures the discrepancy between an estimated causal graph and the ground-truth graph. Let $A, \hat{A} \in \{0,1\}^{d \times d}$ denote the (binary) adjacency matrices of the true and predicted graphs, respectively, where $A_{ij} = 1$ indicates a directed edge $i \to j$. SHD is defined as the minimum number of edge operations required to transform $\hat{A}$ into $A$, where each operation is an *edge insertion*, *edge deletion*, or *edge reversal*. We report lower SHD as better structural accuracy.

In particular, we do not evaluate the causal effect of a variable on itself. Therefore, for all baseline methods, we mask the diagonal entries of the causal matrix and evaluate causal discovery performance only for inter-variable relationships.

## G.4. Normalized Graph Dirichlet Energy for Embedding Smoothness

We employ the graph Dirichlet energy with Rayleigh quotient normalization to quantify the smoothness of learned embeddings over local neighborhoods.

**Graph construction.** Given an embedding matrix $\mathbf{X} \in \mathbb{R}^{n \times d}$ where $n$ is the number of samples and $d$ is the embedding dimension, we first construct a $k$-nearest neighbor ($k$-NN) graph to capture local neighborhood structure. Let $\text{dist}_{ij}$ denote the Euclidean distance between embedding vectors $\mathbf{x}_i$ and $\mathbf{x}_j$. The edge weights are computed using a heat kernel:

$$W_{ij} = \begin{cases} \exp\left(-\frac{\text{dist}_{ij}^2}{2\sigma^2}\right), & \text{if } j \in \mathcal{N}_k(i) \text{ or } i \in \mathcal{N}_k(j), \\ 0, & \text{otherwise,} \end{cases} \tag{42}$$

where $\mathcal{N}_k(i)$ denotes the set of $k$ nearest neighbors of sample $i$, and $\sigma$ is a bandwidth parameter (set to the median of all $k$-NN distances if not specified). The weight matrix $\mathbf{W} \in \mathbb{R}^{n \times n}$ is symmetrized as $\mathbf{W} \leftarrow (\mathbf{W} + \mathbf{W}^\top)/2$.

**Graph Laplacian.** We define the degree matrix $\mathbf{D} = \text{diag}(d_1, \ldots, d_n)$ where $d_i = \sum_{j=1}^{n} W_{ij}$, and construct the unnormalized graph Laplacian:

$$\mathbf{L} = \mathbf{D} - \mathbf{W}. \tag{43}$$

**Dirichlet energy and normalization.** The raw graph Dirichlet energy is computed as:

$$E(\mathbf{X}) = \text{Tr}(\mathbf{X}^\top \mathbf{L} \mathbf{X}) = \frac{1}{2} \sum_{i,j=1}^{n} W_{ij} \|\mathbf{x}_i - \mathbf{x}_j\|^2. \tag{44}$$

To ensure scale invariance (i.e., invariance to global rescaling $\mathbf{X} \mapsto a\mathbf{X}$), we normalize by the Rayleigh quotient:

$$R(\mathbf{X}) = \frac{\text{Tr}(\mathbf{X}^\top \mathbf{L} \mathbf{X})}{\text{Tr}(\mathbf{X}^\top \mathbf{D} \mathbf{X})}. \tag{45}$$

**Interpretation.** The normalized Dirichlet energy $R(\mathbf{X})$ measures the *average squared variation* of embedding coordinates over graph edges, weighted by node degrees. Smaller values of $R(\mathbf{X})$ indicate that nearby points (in the $k$-NN graph) have more similar embedding coordinates, reflecting higher local smoothness. This metric is particularly useful for comparing embeddings across different models or scales, as it is invariant to uniform scaling of the embedding space.

### G.5. Ablation Study

*Table 4.* Ablation study on four datasets.

| Dataset | Metric | Full | -Topo | -Smooth | Rec |
|---------|--------|------|-------|---------|------|
| DP | ROC | **1.000** | 0.875 | 0.750 | 0.875 |
| | PR | **1.000** | 0.833 | 0.750 | 0.833 |
| | SHD | **0.000** | 0.167 | 0.167 | 0.167 |
| UUMC | ROC | **0.853** | 0.760 | 0.787 | 0.773 |
| | PR | **0.714** | 0.477 | 0.498 | 0.491 |
| | SHD | **0.150** | 0.25 | 0.200 | 0.200 |
| fMRI | ROC | **0.949** | 0.938 | 0.937 | 0.934 |
| | PR | **0.744** | 0.596 | 0.642 | 0.551 |
| | SHD | **0.067** | 0.089 | 0.100 | 0.100 |
| CR | ROC | **0.813** | 0.766 | 0.734 | 0.672 |
| | PR | **0.508** | 0.415 | 0.452 | 0.323 |
| | SHD | **0.150** | 0.200 | 0.200 | 0.200 |

## H. Computational Complexity and Efficiency Analysis

Evaluating the computational and memory footprint is essential for practical deployment, especially when highlighting performance gains over standard Time-Delay Embedding (TDE), which is a purely mathematical operation. To provide a comprehensive assessment, we analyze our model's parameter count and conduct hardware benchmarking on the Lorenz-96 dataset (10 variables). The dataset is generated with 2000 time steps and split 7:3 for training and testing. We evaluate both the MLP and TCN variants of our delay embedder against TDE, utilizing an identical CCM pipeline for a fair comparison.

### H.1. Model Complexity and Parameter Count

From a theoretical and architectural standpoint, TopoDistill is a highly lightweight framework.

- **Global Embedder:** Utilizes a 3-layer MLP, accounting for approximately 3k parameters.

- **Global Decoder:** Consists of a single linear layer with fewer than 1k parameters.

- **Delay Embedder:** Configured as a 6-layer MLP (or equivalent TCN), requiring approximately 2k parameters.

In total, the entire neural network framework contains only about 6,000 parameters. This small parameter space inherently limits the memory requirements and prevents severe overfitting.

### H.2. Evaluation of Efficiency

Traditional CCM methods are typically executed on CPUs. To thoroughly benchmark our method, we evaluate it using the PyTorch framework on both a CPU and an NVIDIA RTX 5090 GPU, comparing it against the CPU-based TDE baseline. Table 5 summarizes the execution time (in seconds) and peak memory footprint (in GB) for the entire causal discovery pipeline, which includes training (Stage 1 and Stage 2), embedding generation (inference), and the final CCM scoring.

Because TopoDistill involves neural networks, it inevitably requires more total time than the parameter-free TDE baseline. However, even when executing entirely on a CPU, the total time required by our method is still well under one minute. On a CPU, Stage 1 and Stage 2 training require roughly 20 to 22 seconds combined, maintaining a peak memory footprint of less than 1 GB.

The marginal speedup observed on the GPU is largely due to the fixed I/O overhead and basic scheduling dominating the execution time for such a small parameter space, which further confirms that TopoDistill is highly CPU-friendly. Importantly, once the embedder is trained, the actual inference time for generating embeddings on the test set is extremely short ($< 0.2$ seconds), which is fully comparable to the computational efficiency of TDE.

*Table 5.* Execution time and memory footprint for the entire causal discovery pipeline on the Lorenz-96 dataset. "N/A" indicates that TDE does not require a training phase.

| Metric | TDE (CPU) | Ours-TCN (CPU) | Ours-MLP (CPU) | Ours-TCN (GPU) | Ours-MLP (GPU) |
|---|---|---|---|---|---|
| Stage 1 Train Time (s) | N/A | 1.21 | 1.58 | 1.21 | 2.10 |
| Stage 1 Train Mem (GB) | N/A | 0.70 | 0.70 | 1.31 | 1.30 |
| Stage 2 Train Time (s) | N/A | 18.83 | 20.55 | 5.85 | 10.14 |
| Stage 2 Train Mem (GB) | N/A | 0.87 | 0.84 | 2.03 | 1.90 |
| Embedding Time (s) | 0.29 | 0.18 | 0.07 | 0.18 | 0.20 |
| Embedding Mem (GB) | 0.62 | 0.85 | 0.79 | 2.03 | 1.90 |
| **Total Time (s)** | **22.88** | **54.73** | **56.14** | **41.27** | **46.79** |

## H.3. Performance-Efficiency Trade-off

We acknowledge the fundamental "No Free Lunch" theorem: improving robustness against severe and correlated noise inherently requires an investment of resources. However, based on our benchmarking, we argue that the requirement of approximately 30 additional seconds of one-time training and less than 1 GB of memory is a highly favorable and completely acceptable trade-off for the substantial gains achieved in causal discovery performance.

