# OpenReview forum: "TopoDistill: Distilling Global System Topology for Causal Discovery in Multivariate Time Series"
_ICML.cc/2026/Conference — ICML 2026 regular_

### Official Review · Reviewer_G355 · 2026-02-28

**Soundness:** 4
**Presentation:** 3
**Significance:** 3
**Originality:** 4
**Overall Recommendation:** 5
**Confidence:** 4

**Summary:**

The paper introduces a novel approach to tackle the noise challenge inherent in causality extraction of multi-variate time series.
They propose passing multi-variate of one time step through an encoder-decoder with a next time prediction goal to extract a continuous embedding space (manifold) as a reference space. They then use single variate windowed data through a delay embedder, while using this reference manifold as well, to obtain the shadow manifold.
With leveraging a contrastive loss with early stopping, they ensure the smoothness and the continuity of the obtained shadow manifold. Which is then used with CCM, a KNN distance based scoring approach, to obtain the nearest embedding to the input, hence obtaining Pearson correlation, which in turn helps to understand the causality in the time series.

The authors then utilized multiple causality based datasets, including modern challenging ones such as River Causality, with a number of baselines, to empirically assess the performance of the proposed model.
They then ablate the system, with removing multiple architectures and losses, to better understand the impact of each module.
Finally, they compare TopoDistill with TDE across multiple noise levels and data sizes.

Throughout these results, TopoDistill proved to the most capable model of extracting the causality across the different datasets comapred to other baselines. Moreover, the abaltion results proved the importance of the added modules. Finally, comparing with TDE, highlighted the ability of TopoDistill to withstand increased noise.

**Compliance With Llm Reviewing Policy:**

Affirmed.

**Final Justification:**

My concerns have been fully addressed, which were mainly about the efficiency of the encoder-decoder, which is seen as the main driver behind the increased performance.

**Key Questions For Authors:**

The reviewers need to explain the efficiency aspects of their model, from training time, inference time, memory footprint, and number of parameters. Since the reference embedding manifold is highlighted as a major factor in the enhancement of the results, as highlighted in table-2 and fig-3 and fig-4 by comparing with TDE, this comparison can be seen as a major factor in the assessment.

**Limitations:**

yes

**Strengths And Weaknesses:**

The authors especially appreciate the novelty of using a reference manifold to guide a student shadow manifold is novel, which proves to be able to better extract the causality compared to other models.

Furthermore, for the delay embedder, comparing using MLP or TDE, further justifies that no matter which is the underlying architecture in the delay embedder, the results are mainly driven by how the manifold is guided from reference manifold. This further highlights the novelty of the work.

However, this very point, can as well be seen as somewhat the main weakness of the work. Specifically, since the main driver behind the results is the addition of an encoder-decoder that is trained first before the delay-embedder, main focus on the efficiency (training and inference time/memory) and the number of added parameters should have taken place.
For instance, since it has been mentioned in table-2 that the removal of Topo is an essential ablation factor, and that comparing with TDE is essentially important, as TDE shares identical CCM pipeline, this factor of comparison is extremely essential.

The author would like a detailed discussion on the overhead of the encoder-decoder model that created the reference model, compared to TDE on its own. As since instead of using only one lightweight MLP or TDE encoder in the delay encoder, TopoDistill adds a MLP with multiple layer in the encoder, and a MLP with a single layer to MLP, these might extensively increase the number of parameters.
This discussion will support the claims held by the reviewers, as to justify the architectural increase.

---

> ### Author Rebuttal · Authors · 2026-03-29
>
> Thank you for your comments.
>
> **Response to Weakness \& Question:**
>
> We sincerely thank the reviewer for raising this important point. We fully agree that evaluating the computational and memory footprint is essential for practical deployment, especially when highlighting performance gains over Time-Delay Embedding (TDE), which is a purely mathematical operation.
>
> To provide a comprehensive assessment, we have analyzed our model's parameter count and conducted benchmarking on the Lorenz-96 dataset with 10 variables. The dataset was generated with 2000 time steps and split 7:3 for training and testing. We evaluated both the MLP and TCN variants of our delay embedder against TDE, utilizing the identical CCM pipeline for a fair comparison.
>
> 1. Number of parameters
>
> From a theoretical and architectural standpoint, TopoDistill is a lightweight framework. The global embedder utilizes a 3-layer MLP (\~3k parameters) , the global decoder is a single linear layer (<1k parameters) , and the delay embedder is configured as a 6-layer MLP (\~2k parameters).  **In total, the framework contains only about 6k parameters.**
>
> 2. Evaluation of efficiency
>
> Traditional CCM methods are implemented on CPUs, so we evaluated our method on both CPU and GPU (5090) and compared it with TDE running on CPU. Since our method can use different networks (MLP/TCN) as the embedder, we report results for both architectures separately. “Train-1” and “Train-2” refer to the training processes of the global embedder and the delay embedder, respectively. “Emb” denotes the process of obtaining embeddings (inference) using the delay embedder. “Total Time” refers to the overall time required for the full pipeline, including training, inference, and CCM. The table below summarizes the execution time (s) and memory footprint (GB) for the entire causal discovery pipeline:
>
> |              | TDE(CPU) | Ours-TCN(CPU) | Ours-MLP(CPU) | Ours-TCN(GPU) | Ours-MLP(GPU) |
> |--------------|----------|---------------|---------------|---------------|---------------|
> | Train-1 Time | N/A      | 1.21          | 1.58          | 1.21          | 2.10          |
> | Train-1 Mem  | N/A      | 0.70          | 0.70          | 1.31          | 1.30          |
> | Train-2 Time | N/A      | 18.83         | 20.55         | 5.85          | 10.14         |
> | Train-2 Mem  | N/A      | 0.87          | 0.84          | 2.03          | 1.90          |
> | Emb Time     | 0.29     | 0.18          | 0.07          | 0.18          | 0.20          |
> | Emb Mem      | 0.62     | 0.85          | 0.79          | 2.03          | 1.90          |
> | Total Time   | 22.88    | 54.73         | 56.14         | 41.27         | 46.79         |
>
> Since neural networks require training, TopoDistill inevitably takes more total time than TDE. However, even when running on a CPU, the total time required by our method is still under 1 minute. **On a CPU, Stage 1 and Stage 2 training require roughly 20 seconds combined, with a peak memory footprint of less than 1 GB**. The marginal speedup on the GPU is largely due to I/O overhead and basic scheduling for such a small parameter space, further confirming that TopoDistill is highly CPU-friendly. Once the embedder is trained, the actual inference time for generating embeddings on the test set is very short (< 0.2 s), comparable to the efficiency of TDE.
>
> 3. Performance-efficiency trade-off
>
> We acknowledge the fundamental "No Free Lunch" theorem: improving robustness against severe noise inherently requires an investment. However, we argue that **an additional ~30 seconds of one-time training and <1 GB of memory** is a favorable, completely acceptable trade-off for the gains in causal discovery performance.
>
> We will include this detailed efficiency analysis and the corresponding table in the Appendix of the revised manuscript, and reference it in the main text to provide readers with a transparent view of the model's computational footprint.

---

> > ### Author Rebuttal · Reviewer_G355 · 2026-04-03
> >
> > I would like to thank the authors for their discussion, the clarification as you have done is essential to understand the impacts of the encoder-decoder on the added architecture.
> > My concerns have been fully addressed, just make sure to include this discussion in the final manuscript. (I have raised the score)

---

> > > ### Author Response · Authors · 2026-04-04
> > >
> > > Thank you very much for your positive feedback and for raising the score. We are truly glad that our clarification addressed your concerns. We will make sure to include this discussion in the final manuscript. We sincerely appreciate your time and effort.

---

### Official Review · Reviewer_U291 · 2026-03-05

**Soundness:** 3
**Presentation:** 3
**Significance:** 3
**Originality:** 3
**Overall Recommendation:** 4
**Confidence:** 2

**Summary:**

This work addresses the challenge of causal discovery from noisy multivariate time series, where traditional methods like Convergent Cross Mapping (CCM) rely on time-delay embedding to reconstruct system dynamics from a single noisy signal, often leading to distorted manifolds and unreliable neighborhood structures.
The authors propose a topology informed framework that improves univariate shadow-manifold reconstruction by aligning it with a multivariate global representation of the system's attractor.
A global embedder learns the overall topology from multivariate data, while a delay embedder is trained to match its neighborhood structure, producing smoother and more reliable embeddings.
The authors provide both theoretical analysis and experiments on data.

**Compliance With Llm Reviewing Policy:**

Affirmed.

**Final Justification:**

The authors have addressed my concerns. I find the article an interesting take on causality from a "deterministic" perspective.
Even though I personally find more interesting and applicable the notion of causality "a la Pearl," the authors' contribution seems a theoretically solid scientific contribution which might deserve acceptance to the conference.

**Key Questions For Authors:**

1) Can the authors explain what they mean by separability assumptions for Granger causality?
Many variations of Granger causality exist that take into account nonlinear dynamics.
Also the concept of Granger causality has been extended/modified using for example directional entropy.
2) Could the authors explain how their work might connect to the notion of causality a la Pearl? In this respect, could their one-step-ahead predictors for the training be replaced by predictors that can estimate variables at the same time (in a way similar to Dimovska, "A Control Theoretic Look at Granger Causality: Extending Topology Reconstruction to Networks With Direct Feedthroughs")? Would this allow one to detect "instantaneous" causation?

**Limitations:**

yes

**Strengths And Weaknesses:**

Strengths

- S1) The authors provide convincing a well though approach to improve the performance of CCM by using a latent space that should preserve the topology of the original system

Weaknesses

- W1) Being still based on Taken's emmbedding, the causal detection might be still sensitive to noise despite the authors' efforts to mitigate its effects.
- W2) The method cannot detect "instantaneous causality" (causality with no lags)
- W3) The sample complexity of the method might be high

---

> ### Author Rebuttal · Authors · 2026-03-30
>
> **Response to Weakness 1:**
>
> We thank the reviewer for this critical observation. While standard Time-Delay Embedding (TDE) rigidly propagates measurement noise into the embedding space , TopoDistill inherently circumvents this by shifting to a learnable neural embedding framework.
>
> Our noise mitigation is structurally guaranteed by both theory and empirical design:
>
> Theoretical Guarantee: The global embedder is trained on a one-step-ahead prediction task. This objective naturally drives the encoder to **extract features that maximize information about future states while actively suppressing noise components, which are uninformative about future evolution.** As formalized in Theorem C.1 (Noise-Robust Homeomorphism), our neural embedder theoretically maintains an $\mathcal{O}(\delta)$-homeomorphism , effectively neutralizing normal noise directions. This denoised topology is then distilled to the delay embedder.
>
> Empirical Robustness: We explicitly stress-tested our method on the Lorenz-96 dataset with extreme Gaussian noise (up to std=2.0, approx. 46\%). Figure 3b demonstrates that TopoDistill maintains a remarkably low Dirichlet Energy (indicating a smooth manifold) where standard TDE fails. Consequently, Figure 4 shows TopoDistill achieves stable convergence plateaus for causal discovery even under these severe noise regimes.
>
> We will clarify this fundamental distinction between our learnable embedding and rigid TDE in the revised manuscript.
>
> **Response to Question 1:**
>
> The "separability assumption" refers to the premise that **causal effects can be decomposed into independent contributions.** Granger causality (GC)-based frameworks fundamentally test for causality by assessing whether the predictability of an effect decreases when the candidate cause is removed from the model.
>
> While advanced GC variations are powerful, **this core assumption is often fundamentally violated in tightly coupled nonlinear systems.** Because these variables are linked on the same manifold, the historical information of the effect variable completely encodes the state of its cause. Consequently, mathematically removing the cause often does not decrease predictability, leading to potential false negatives in GC-based frameworks.
>
> CCM offers a complementary approach precisely because it does not rely on this separability assumption. It tests whether the cause can be reconstructed from the effect's shadow manifold, rather than attempting to separate their independent contributions. We will add a description of this point in the Appendix.
>
> **Response to Weakness 2 \& Question 2:**
>
> We deeply appreciate these profound theoretical questions, which highlight important boundaries of our current framework.
>
> 1. Connection to Pearl’s Framework: Pearl’s framework is fundamentally rooted in probabilistic graphical models (DAGs) and interventions (do-calculus) for stochastic systems. In contrast, CCM and TopoDistill are grounded in deterministic dynamical systems and topology (Takens’ theorem). While Pearl’s approach models stochastic conditional independence, our method leverages deterministic state-space reconstruction. They represent complementary paradigms: one excels in probabilistic interventions, while ours resolves tightly coupled deterministic dynamics where standard interventions may be unfeasible.
>
> 2. Instantaneous Causality \& Direct Feedthroughs: The reviewer is correct (Weakness 2) that standard CCM, and currently TopoDistill, assumes a finite propagation delay, meaning instantaneous causality is out of scope. However, your suggestion regarding Dimovska's control-theoretic direct feedthroughs is exceptionally insightful. Conceptually, we could augment our global embedder’s one-step-ahead decoder with a direct feedthrough matrix to allow specific components of $x\_t$ to instantly predict other components of $x\_t$. Implementing this while maintaining the global topological constraints presents a highly viable path to extending our framework for instantaneous causation.
>
> **Response to Weakness 3:**
>
> We completely agree with the reviewer’s intuition. Because TopoDistill relies on training neural networks to learn topological mappings, it inherently requires more data than non-parametric baselines like standard TDE. We acknowledged this as a primary limitation in the Conclusion section.
>
> However, we view this higher sample complexity as a necessary and worthwhile trade-off. It is the cost paid to achieve the significant robustness against severe noise and complex nonlinearities, which rigid linear embeddings cannot handle, as evidenced by our results in Figure 3.
>
> While the sample complexity is relatively higher, our data requirements remain practical for typical real-world scenarios. Furthermore, as noted in our Conclusion, incorporating few-shot learning techniques is a highly promising direction to further mitigate this requirement in future work.

---

> > ### Author Rebuttal · Reviewer_U291 · 2026-04-01
> >
> > I thanks the authors for their answers. I considered all my questions substantially answered.

---

> > > ### Author Response · Authors · 2026-04-03
> > >
> > > Thank you very much for your positive feedback. We are glad that our responses have addressed all your concerns. We sincerely appreciate your time and effort in reviewing our work.

---

### Official Review · Reviewer_xcAF · 2026-03-10

**Soundness:** 2
**Presentation:** 1
**Significance:** 2
**Originality:** 3
**Overall Recommendation:** 4
**Confidence:** 4

**Summary:**

This paper proposes TopoDistill, a topology-informed knowledge distillation framework for causal discovery in multivariate time series. The method aims to improve the robustness of Convergent Cross Mapping (CCM) under noisy observations by learning a multivariate “global” embedding of the system and distilling its neighborhood topology into univariate delay embeddings used for shadow manifold reconstruction. The resulting embeddings are then used for CCM-based causal inference.

While the problem is interesting and the paper is generally well written, I have several concerns regarding the theoretical justification and the positioning with respect to prior causal discovery literature, which limit my confidence in the contributions.

**Compliance With Llm Reviewing Policy:**

Affirmed.

**Final Justification:**

The authors have addressed most of my concerns regarding presentation and the mechanism behind TopoDistill.

**Key Questions For Authors:**

The theoretical results assume specific properties of the learned embedding function (e.g., tangential preservation and normal compression). How are these properties encouraged or guaranteed by the proposed training objective?

How does the proposed approach relate to transfer entropy and other information-theoretic causal discovery methods?
Could the authors clarify the conceptual differences and potential advantages?

Would the proposed representation learning strategy also improve non-CCM causal discovery methods, or is the benefit specific to CCM-style reconstruction-based inference?

**Limitations:**

Limited explanation of why the method works （loss--->topo, why?）

While the paper provides theoretical motivation for the proposed topology distillation framework, the mechanism explaining *why* the method should improve causal discovery remains unclear.

The theoretical section relies on assumptions about the learned encoder and approximate manifold homeomorphism, but it is not evident how the actual optimization objective enforces these properties in practice. As a result, the connection between the theoretical arguments and the empirical effectiveness of the method remains somewhat indirect.

In particular, the paper introduces the notion of "topology distillation", but in the implementation this topology appears to be derived directly from encoder-produced representations. It is therefore unclear:

- what specific topological properties are being preserved,
- how the distillation objective guarantees such properties,
- and why this procedure should systematically improve the quality of shadow manifold reconstruction used by CCM.

Overall, the paper would benefit from a clearer explanation of the mechanism behind the method and a more explicit connection between the theoretical motivation and the actual implementation.

**Strengths And Weaknesses:**

## Strengths
The paper addresses the robustness issue of delay embedding under noisy observations.

The proposed topology distillation idea is intuitive and empirically improves CCM performance in several experiments.

## Weakness
1. Limited theoretical justification.
The theoretical analysis relies on strong assumptions about the learned encoder (e.g., tangential preservation and normal compression), but these properties are not explicitly enforced by the training objective. As a result, the theoretical results provide only heuristic intuition rather than rigorous guarantees for the proposed method.

2. Insufficient discussion of related work.
The paper is not well positioned with respect to prior literature on information-theoretic causal discovery, particularly approaches based on Transfer Entropy and related information flow measures. Since CCM-style reconstruction and transfer entropy methods both aim to capture causal influence via predictive information in time series, the connection should be discussed and compared.
### 3. Clarity and notation issues in the presentation

Several formulas and notations in the paper are not clearly defined or appear inconsistent. For example:

- **Page 3:** The notation of \(F_X\) is unclear, as it is not consistent whether \(X\) denotes a scalar variable or a vector-valued state.

- **Page 4:** The sentence following one of the equations appears to contain a missing parenthesis, which makes the expression difficult to parse.

- **Eq. (6):** The meaning of the weight \(w\) is not clearly defined in the text.

- **Eq. (14):** The expression \(\lambda + (1-\lambda)\) appears in the formulation without a clear explanation of its role in the objective.

These issues make the method harder to follow and suggest that the paper would benefit from a more careful and consistent presentation.

---

> ### Author Rebuttal · Authors · 2026-03-26
>
> Thank you for your comments.
>
> **Response to Weakness 1, Question 1 \& Limitations:**
>
> We sincerely thank the reviewer for this critique. While avoiding computationally expensive explicit Jacobian regularization, our objectives implicitly enforce these properties.
>
> 1. What specific topological properties are preserved?
>
> The preserved "topology" refers to **the system's local neighborhood structure** under diffeomorphism. It means maintaining relative point ordering within a local neighborhood to reflect true dynamical proximity rather than noise-induced spurious proximity.
>
> 2. How do the objectives enforce these properties?
>
> Normal Compression ($\mathcal{L}\_{global}$ \& $\mathcal{L}\_{topo}$): The global embedder's prediction loss ($\mathcal{L}\_{global}$) naturally filters out unpredictable measurement noise, creating a denoised topological template. By minimizing the KL-divergence ($\mathcal{L}\_{topo}$) against this implicitly denoised teacher distribution, **the delay embedder is heavily penalized if it expands in normal (noise) directions**. This effectively enforces normal compression without explicit regularization.
>
> Tangential Preservation ($\mathcal{L}\_{smooth}$): Our temporal contrastive loss explicitly pushes temporally distant negative samples apart. This negative sampling prevents representation collapse and stretches the temporal trajectory, naturally maintaining the required positive lower bound for the tangential derivative to preserve the signal.
>
> 3. Why does this systematically improve CCM?
>
> CCM relies fundamentally on Euclidean k-nearest neighbors. **TopoDistill ensures Euclidean proximity equates to true dynamical similarity**, eliminating spurious noise-induced neighbors.
>
> We will add a "Mechanism" paragraph in Section 4 outlining this pipeline, and expand Appendix C.1 to formalize how these losses act as implicit regularizers.
>
> **Response to Weakness 2 \& Question 2:**
>
> We agree that explicitly contrasting our method with information-theoretic approaches, such as Transfer Entropy (TE), clarifies our contribution.
>
> While both TE and CCM-based methods leverage predictability for causal discovery, their conceptual foundations and operational mechanisms fundamentally differ:
>
> 1. Conceptual Foundation: TE is grounded in Shannon information theory and measures uncertainty reduction forward in time. It inherently assumes "separability" (similar to Granger Causality). In contrast, TopoDistill builds on dynamical systems theory (Takens' theorem). It identifies causality by exploiting the topological entanglement of variables in non-separable, deterministically coupled systems, predicting the cause from the effect's reconstructed history.
>
> 2. Potential Advantages: TE requires estimating high-dimensional joint probability density functions, which is computationally expensive and highly sensitive to measurement noise in continuous time series. TopoDistill bypasses complex density estimation entirely. By extracting a low-dimensional, noise-filtered topological template via $\mathcal{L}\_{global}$ (Appendix A), our framework geometrically aligns the shadow manifolds, making it significantly more robust to noise and high-dimensional observational challenges.
>
> We will add a dedicated discussion to Section 2 in the revision. to better position TopoDistill within the broader causal discovery landscape.
>
> **Response to Weakness 3:**
>
> We sincerely apologize for the confusion. We have carefully revised the manuscript to address all the issues you raised. Here, we briefly clarify a key point:
> In our original draft, $w\_t^{(i)}$ was used to denote **a temporal "window,"** but we agree this easily conflicts with standard notation for network "weights." We have replaced this notation throughout the text. Eq. (6) now uses standard slice notation: $z\_{t}^{(i)}=f\_{s}^{(i)}(x\_{t-\tau+1:t}^{(i)})$, and we immediately define $x\_{t-\tau+1:t}^{(i)}$ as the temporal window of length $\tau$ directly below the equation.
>
> **Response to Question 3:**
>
> Our strategy is tailored to CCM for causal discovery. TopoDistill optimizes geometric fidelity and neighborhood smoothness, which CCM explicitly exploits via distance-based cross-mapping. Conversely, statistical methods like Granger causality require variable separability. Distilling a global template deliberately entangles univariate embeddings to reconstruct the global attractor. This violates their separability assumptions and risks inflating false positives.
>
> Beyond causal discovery, however, this representation learning paradigm holds significant promise for broader time series tasks. For univariate forecasting or missing data imputation in coupled systems, distilling global multivariate topology into a single-sensor embedder allows models to implicitly leverage robust, system-level dynamics even when only partial or noisy univariate observations are available at inference time. We will clarify this broader scope and potential in the revised version.

---

> > ### Author Rebuttal · Reviewer_xcAF · 2026-04-03
> >
> > The authors have addressed most of my concerns regarding presentation and the mechanism behind TopoDistill.

---

> > > ### Author Response · Authors · 2026-04-03
> > >
> > > Thank you very much for your positive feedback and for raising your score. We are truly glad to hear that our revisions have addressed most of your concerns. Your insightful comments have been very helpful in improving our work.
> > >
> > > Once again, thank you for your time and valuable input.

---

### Official Review · Reviewer_UcUq · 2026-03-12

**Soundness:** 4
**Presentation:** 3
**Significance:** 4
**Originality:** 3
**Overall Recommendation:** 5
**Confidence:** 3

**Summary:**

This paper introduces TopoDistill, a framework that enhances noise-robust causal discovery in multivariate time series by refining Convergent Cross Mapping (CCM). It employs a topology-informed knowledge distillation approach, where a global teacher captures system-level dynamics to guide a delay-based student in reconstructing denoised univariate manifolds. The framework's effectiveness is validated across synthetic Lorenz-96 systems and real-world benchmarks like fMRI and Causal Rivers, demonstrating superior robustness and accuracy compared to recent deep-learning-based causal discovery methods.

**Compliance With Llm Reviewing Policy:**

Affirmed.

**Final Justification:**

After considering both the paper and the rebuttal, I raise my final recommendation to Accept. Overall, I find the paper technically sound, reasonably original, and clearly presented.

My main concerns in the original review were about robustness under temporally correlated noise, the early stopping criterion, and the method's positioning relative to related work. The rebuttal addressed these concerns well. In particular, the additional AR(1) noise experiments and explanation provided a convincing response to my primary concern about robustness under correlated noise. The clarifications on early stopping, computational cost, and applicability under non-stationarity were also helpful.

Overall, the rebuttal reinforced and strengthened my initial positive assessment. It addressed my main concerns and justifies raising my score to 5.

**Key Questions For Authors:**

1. How does the "implicit denoising" mechanism perform under correlated noise scenarios (e.g., AR(1) noise) where the noise component exhibits its own temporal dependency?
2. What theoretical or empirical evidence ensures that the "manifold smoothness" stopping criterion does not lead to information collapse or the loss of micro-scale attractor structures?
3. In the Causal Rivers experiment, what were the specific computational requirements for the distillation phase relative to the final cross-mapping step?
4. How do you handle cases where the system attractor is non-stationary and its topology changes over time?

**Limitations:**

yes

**Strengths And Weaknesses:**

Strengths:
- Robustness to Observational Noise: The framework effectively utilizes system-level information as a supervisory signal to rectify local-view distortions. This approach strengthens the reliability of distance-based predictions in CCM [1] without violating the requirement that causal inference must proceed from univariate historical reconstructions.
- Integration of Dynamical Systems and Representation Learning: The transition from rigid, linear TDE to learnable embeddings is well-motivated by Takens' theorem. The use of KL divergence for aligning neighborhood distributions provides a systematic way to preserve local isometry in the presence of noise.
- Empirical Validation on Diverse Benchmarks: The model demonstrates competitive performance against recent baselines such as CUTS+ and AERCA [3]. The inclusion of the Causal Rivers dataset provides meaningful evidence of the framework's applicability to complex, real-world environmental monitoring tasks.

Weaknesses:
- Strong Assumption on Noise Independence: The framework relies on the assumption in Sec 4.1 that measurement noise is independent of future states to achieve "implicit denoising." However, in real-world systems like Causal Rivers, noise is often temporally correlated (e.g., red noise). If the noise itself follows a predictable pattern, the global teacher might inadvertently model the noise components, thereby transferring noise-induced correlations to the student embedder rather than filtering them out.
- Potential Instability of the Smoothness-based Early Stopping: The paper proposes an unsupervised early-stopping criterion based on manifold smoothness. While motivated by the need to prevent overfitting to noise, there is no formal guarantee that this metric won't favor degenerate mappings (e.g., collapsing the manifold into a simpler geometric form that loses critical chaotic dynamics). The lack of sensitivity analysis regarding how this threshold varies across different dynamical regimes poses a risk to the model's practical generalizability.
- Comparison with Contemporary Multivariate CCM Variants: While the paper cites MXMap [2] in the related work, it lacks a direct empirical comparison with this specific paradigm. Since MXMap [2] also addresses multivariate causal discovery under complex conditions via structural pruning, a quantitative comparison would better establish the unique contribution and relative performance of TopoDistill's distillation approach.


[1] Sugihara, George, et al. "Detecting causality in complex ecosystems." science 338.6106 (2012): 496-500.

[2] Zhang, Elise, et al. "MXMap: A Multivariate Cross Mapping Framework for Causal Discovery in Dynamical Systems." arXiv preprint arXiv:2502.03802 (2025).

[3] Han, Xiao, et al. "Root cause analysis of anomalies in multivariate time series through granger causal discovery." The Thirteenth International Conference on Learning Representations. 2025.

---

> ### Author Rebuttal · Authors · 2026-03-30
>
> Thank you for your comments.
>
> **Response to Weakness 1 \& question 1**:
>
> We sincerely thank the Reviewer for this highly insightful question. Prompted by your intuition, we evaluated TopoDistill on a 10-variable Lorenz-96 system with 40% AR(1) noise ($n\_t = \rho n\_{t-1} + \epsilon\_t$).
>
> Varying the autocorrelation $\rho$ revealed a surprising inverted-U performance curve:
>
> $\rho=0.0$: ROC $0.969$ / PR $0.934$ (White Noise)
>
> $\rho = 0.2$: ROC $0.974$ / PR $0.951$
>
> $\rho=0.4$: ROC $0.988$ / PR $0.980$ (Peak Performance)
>
> $\rho = 0.6$: ROC $0.988$ / PR $0.975$
>
> $\rho=0.8$: ROC $0.980$ / PR $0.946$ (Extreme Red Noise)
>
> This phenomenon illustrates the interplay between noise spectral properties and TopoDistill’s inductive biases:
>
> 1. Benefit of Moderate Correlation ($\rho \in [0.2, 0.6]$):
>
> Moderate AR(1) acts as a low-pass filter. Unlike erratic white noise, this temporally "smoother" perturbation aligns better with our $\mathcal{L}\_{smooth}$ (tangential preservation) prior. It makes decoupling the noise from the continuous deterministic manifold structurally easier.
>
> 2. Risk of Extreme Correlation ($\rho \ge 0.8$):
>
> Exactly as you hypothesized, highly correlated noise forms a persistent random walk (a spurious dynamic). The global teacher models this, which then competes for the student's capacity bottleneck (~6k parameters), causing the slight performance dip.
>
> Importantly, even at $\rho=0.8$, TopoDistill outperforms the pure white noise baseline. This confirms that while your theoretical concern manifests under extreme conditions, our topological constraints successfully force the student to prioritize the dominant deterministic causal manifold over stochastic fluctuations. We are extremely grateful for this comment, and we will add these results to the Appendix.
>
> **Response to Weakness 2 \& question 2**:
>
> We guarantee against information collapse both theoretically and empirically:
>
> 1. Theoretical Guarantee:
>
> The early-stopping metric $S\_m$ is **purely a passive monitor**, not the optimization objective. The actual geometric constraint is the contrastive loss $\mathcal{L}\_{smooth}$. Crucially, its **InfoNCE negative-sampling denominator actively pushes temporally distant states apart**, mathematically preventing the embeddings from collapsing into a degenerate point or overly simple shape.
>
> 2. Empirical Evidence:
>
> To prove $S\_m$ prevents noise-fitting without causing collapse, we analyzed training dynamics on Lorenz-96 with noise. While the total loss strictly decreases ($8.01 \rightarrow 4.52$), $S\_m$ follows a V-shaped curve ($0.0089 \rightarrow 0.0041 \rightarrow 0.0042$ (early stopped) ).
>
> $S\_m$ firstly drops, because the student accurately learns the deterministic causal manifold, smoothing out high-frequency measurement noise. Then $S\_m$ rebounds. Due to the lack of a full multivariate view of the teacher, the univariate student eventually begins to overfit local noise to forcefully minimize $\mathcal{L}\_{topo}$. This noise memorization reintroduces trajectory jitter, causing $S\_m$ to increase.
>
> Therefore, early stopping at the $S\_m$ inflection point helps mitigate severe noise overfitting. We will add this analysis to the Appendix.
>
> **Response to Weakness 3:**
>
> We omitted a quantitative comparison due to fundamental incompatibilities. Methodologically, TopoDistill’s representation enhancement and MXMap’s structural pruning are orthogonal. Evaluation-wise, MXMap’s dual-phase hard thresholds preclude threshold-free metrics, and its equating of non-oriented with bidirectional edges conflicts with our strict directed-graph assessment. Rather than forcing an unfair threshold-sensitive comparison, we will  add a principled discussion in the Appendix.
>
> **Response to Question 3:**
>
> The distillation stage represents a one-time training cost. We follow the data processing pipeline released by the dataset authors. For the Causal Rivers dataset, we did not require any specialized algorithmic optimizations or custom hardware beyond our standard model setup. We conducted a detailed computational overhead analysis on the Lorenz-96 dataset (10 variables, 2000 time points). The results show that, even when running on a CPU, the total training time is less than 30 seconds (detailed results are in response to RW G355).
>
> **Response to Question 4:**
>
> While standard CCM assumes a stationary attractor, TopoDistill addresses non-stationarity from two perspectives:
>
> 1. Empirically, TopoDistill already demonstrates strong robustness on datasets with inherent non-stationary dynamics and seasonal patterns (e.g., fMRI, Causal Rivers), consistently outperforming baselines.
>
> 2. Promising extension: For systems experiencing severe or abrupt topological shifts, assuming piecewise stationarity is necessary. Because TopoDistill is computationally lightweight (requiring only <30s for the full training pipeline and <0.2s for inference), dynamically retraining the embedders over localized sliding windows is highly feasible.

---

> > ### Author Rebuttal · Reviewer_UcUq · 2026-04-03
> >
> > The rebuttal fully addressed my main concerns. In particular, the additional AR(1) noise experiments and the accompanying explanation provide a convincing answer to my question about robustness under temporally correlated noise.
> >
> > The clarification on the smoothness-based early stopping criterion is also helpful and resolves my concern about potential collapse or loss of fine-scale structure. The responses on computational cost, MXMap, and non-stationarity are clear as well.
> >
> > Overall, I find the rebuttal convincing, and will raise my score.

---

> > > ### Author Response · Authors · 2026-04-03
> > >
> > > Thank you very much for your thoughtful and encouraging feedback. We are truly glad that our rebuttal addressed your concerns. Your insightful comments have helped us strengthen the paper, and we greatly appreciate your time and effort.

---

### Decision · Program_Chairs · 2026-04-30

**Decision:**

Accept (regular)

**Comment:**

This submission introduces TopoDistill, i.e., a topology-informed distillation framework that improves (CCM-style) causal discovery from noisy multivariate time series. All reviewers found the core idea novel, well motivated, and empirically strong across synthetic and real benchmarks. The concerns raised were mostly about mechanism, positioning, correlated (AR(1)) noise, and efficiency. Also, the rebuttal addressed those issues with specific clarifications and additional evidence. The main remaining weakness, from my perspective, is that some of the theoretical language still looks more like informed justification than a tight guarantee, and the framing relative to broader causal-discovery questions remains somewhat narrow. Nevertheless, from my assessment, this looks more like calibration and exposition issues, not fundamental issues which would require another round of reviews. Overall, given that all reviewers ended positive, and several explicitly raised their scores after rebuttal, and the remaining issues seem addressable with modest revision (i.e., tightening wording, scope, and adding clarifications made during the rebuttal phase), I am recommending an "Accept" here.